# Generating Code World Models with Large Language Models Guided by Monte Carlo Tree Search

**Nicola Dainese**\*
Department of Computer Science
Aalto University
nicola.dainese@aalto.fi

**Matteo Merler**\*
Department of Computer Science
Aalto University
matteo.merler@aalto.fi

**Minttu Alakuijala**
Department of Computer Science
Aalto University
minttu.alakuijala@aalto.fi

**Pekka Marttinen**
Department of Computer Science
Aalto University
pekka.marttinen@aalto.fi

## Abstract

In this work we consider Code World Models, world models generated by a Large Language Model (LLM) in the form of Python code for model-based Reinforcement Learning (RL). Calling code instead of LLMs for planning has potential to be more precise, reliable, interpretable, and extremely efficient. However, writing appropriate Code World Models requires the ability to understand complex instructions, to generate exact code with non-trivial logic and to self-debug a long program with feedback from unit tests and environment trajectories. To address these challenges, we propose Generate, Improve and Fix with Monte Carlo Tree Search (GIF-MCTS), a new code generation strategy for LLMs. To test our approach in an offline RL setting, we introduce the Code World Models Benchmark (CWMB), a suite of program synthesis and planning tasks comprised of 18 diverse RL environments paired with corresponding textual descriptions and curated trajectories. GIF-MCTS surpasses all baselines on the CWMB and two other benchmarks, and we show that the Code World Models synthesized with it can be successfully used for planning, resulting in model-based RL agents with greatly improved sample efficiency and inference speed.

## 1 Introduction

The ability to model the world is essential for goal-oriented intelligent agents [Ha and Schmidhuber, 2018]. When faced with a novel environment, the agent must quickly understand its mechanics to achieve its goal, for example by building an internal representation of the world and planning with it. In this context, natural language conditioning can be useful for grounding current observations in past knowledge and improving the agent's understanding of the world. Therefore, communicating information about a new task to the agent in natural language is particularly promising, and multiple works explore instruction-following agents [Jang et al., 2022, Ahn et al., 2022]. However, not all important information can be communicated in the form of imperative instructions. Many key facts required to solve a task involve understanding observations, predicting outcomes of different actions and determining whether those outcomes align with the agent's goals. Thus, systems capable of leveraging additional descriptive information, such as model-based Reinforcement Learning (RL) agents, have a greater potential for fast and efficient adaptation via natural language [Lin et al., 2024].

---

\*Asterisk indicates equal contribution.

38th Conference on Neural Information Processing Systems (NeurIPS 2024).

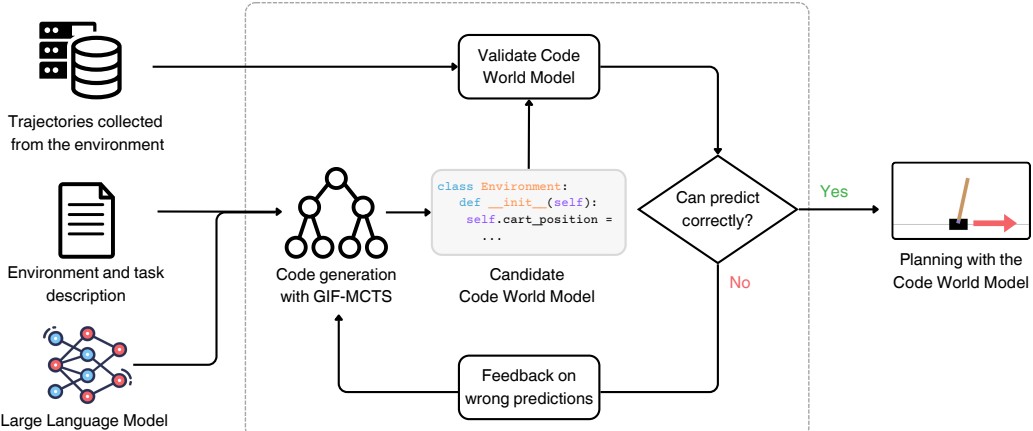

Figure 1: Overview of the Code World Models (CWM) framework. Given the description of an environment and a task, we use an LLM guided by the GIF-MCTS method to iteratively generate and refine a candidate CWM. The candidate's correctness is evaluated by checking if it correctly predicts a set of trajectories collected from the true environment. If the model cannot fully predict all transitions, the fraction of correct predictions and other information are given as feedback to the LLM and the cycle repeats. After matching all transitions or having used up a computational budget, the best CWM is returned and used to solve the task via model-based planning.

Large Language Models (LLMs) have revolutionized the field of Natural Language Processing, and offer great opportunities for world modeling, thanks to their internet-scale knowledge, reasoning, and instruction-following abilities. However, it is not clear how to best combine LLMs and world models. One option is multi-modal systems such as text-to-video models [Gupta et al., 2023], which present the highest prediction fidelity, language understanding and out-of-distribution generalization for generation tasks, yet they are too slow to be called repeatedly in a planning loop due to their high inference cost. On the other hand, language-conditioned model-based RL agents [Dainese et al., 2023, Lin et al., 2024] are typically fast at planning and easily trainable. However, they cannot conveniently incorporate LLMs because of their specialised architectures and as such have poor language understanding and generalization capabilities. Other works, such as [Hao et al., 2023], perform planning using an LLM as a world model directly, but they are slow for inference and restricted to textual inputs and outputs, limiting their applicability in RL.

In this study we propose to model the world with code, rather than directly predicting the future with an LLM, which is known to be costly, slow and unreliable. In contrast, code is precise, fast, reliable and interpretable. We thus introduce Code World Models (CWMs), a novel approach to generate RL world models by writing Python code with an LLM, for which a high-level overview can be seen in Figure 1. The concept of CWMs has been independently and contemporaneously proposed by Tang et al. [2024b]; however, our method is technically distinct (Section 2) and scales to more complex world models (Section 5). Alongside this paradigm, we introduce the Code World Models Benchmark (CWMB), consisting of 18 diverse RL environments for discrete and continuous control, paired with corresponding natural language descriptions and curated trajectories. This benchmark aims to facilitate the accurate synthesis of Code World Models through learning from the provided data and evaluate different code generation methods across environments of varying complexity.

Synthesizing programs for world models requires complex reasoning, precise instruction following, accurate implementation of the environment dynamics and reward functions, as well as coding skills for debugging and refining long programs using unit tests. To meet these challenges we propose Generate, Improve and Fix with Monte Carlo Tree Search (GIF-MCTS), a new code generation method based on Monte Carlo Tree Search (MCTS, Kocsis and Szepesvári [2006]) for LLMs, especially suited for generating Code World Models.[2] We evaluate the performance of our method on three benchmarks: the new CWMB, the *Competition* split on APPS [Hendrycks et al., 2021], a popular and challenging coding benchmark, and RTFM [Zhong et al., 2020], a language-conditioned grid-world, showcasing environments with varying characteristics and complexity. GIF-MCTS

---

[2]We release our code at https://github.com/nicoladainese96/code-world-models.

outperforms existing methods on all three benchmarks. Moreover, we demonstrate successful planning in several environments using the synthesized CWMs. This results in model-based RL agents with exceptional sample efficiency and inference speed (from four to six orders of magnitude faster compared to directly querying an LLM as a world model, as shown in Appendix H), while, provided the CWM is accurate, matching the performance of an oracle planner with access to the real-world model. Finally, we discuss the limitations and challenges to overcome to make Code World Models more broadly applicable.

## 2 Related Work

**World models with code.** Code is a promising choice for predictive world models thanks to its fast inference, exact syntax and interpretable behavior. However, code alone often struggles to cover the entire scope of the environment's dynamics and previous works often uses different techniques to build a full world model. AutumnSynth [Das et al., 2021] uses a custom programming language named Autumn and integrates a functional synthesis step with a synthesized finite-state automata to model any latent variable. Another popular choice is the Planning Domain Definition Language (PDDL) [Ghallab et al., 1998], which expresses actions as a set of preconditions and effects on the environment. However, PDDL approaches, as in the works by Guan et al. [2023] and Wong et al. [2024], are reliant on having access to predicates about the environment and plan in terms of high-level language actions, which need a low-level language-conditioned controller to be carried out. LLMs have also been used to generate a model based on probabilistic code [Wong et al., 2023].

Most similar to our approach, the concurrently proposed WorldCoder[3] [Tang et al., 2024b] also leverages LLMs to generate a Python-based world model. WorldCoder chooses a program to refine from a working set of programs using the classical Thompson Sampling bandit algorithm [Thompson, 1933, Katehakis and Veinott, 1987], informed by a Beta prior, to iteratively learn a world model from gathered experience. Tang et al. focus on learning world models from online interactions with the environment in two grid-world tasks and on transferring knowledge across variants of the same task. We instead consider a broader selection of environments, propose to learn from offline data, and handle continuous state and action spaces in addition to discrete worlds. Furthermore, we rigorously benchmark and ablate our code generation method, GIF-MCTS, achieving state-of-the-art results on the *Competition* split of the APPS coding benchmark, and obtain superior or on par performance to WorldCoder on CWMB.

**Code generation with LLMs.** Current state-of-the-art code generation methods all employ LLMs. While improvements to this task can come from both advancements in the LLMs' coding abilities and enhancements in prompting strategies to guide LLM decoding, the latter is the most relevant to our work. A host of prompting techniques have shown how to leverage the In-Context Learning (ICL) [Brown et al., 2020] abilities of LLMs to enhance a model's reasoning skills, and, as a result, the quality of generated programs. Perhaps the most influential of these is Chain of Thought (CoT) [Wei et al., 2022, Kojima et al., 2022], which leverages in-context examples to encourage intermediate reasoning steps. Tree-like approaches based on the CoT method have also been presented [Yao et al., 2023, Hao et al., 2023]. The work by Zhang et al. [2023] proposes to guide the LLM generation with an MCTS method based on the feedback from unit tests. However, the method considers every token decoded by the LLM as an action in the MCTS tree, which becomes impractical when we have hundreds of tokens per program.

Most similar to our method, LATS [Zhou et al., 2023] uses an MCTS-based generation strategy that incorporates both self-reflection [Madaan et al., 2023, Shinn et al., 2023, Gou et al., 2024] and feedback from the environment. While LATS is broadly applicable to reasoning tasks, it has limitations in code-specific applications like ours. For instance, it generates $n$ programs simultaneously from the same node, rather than sequentially, which does not fully exploit the sequential nature of MCTS. Additionally, it uses a separate prompt to reflect on incorrect code predictions, whereas we integrate self-reflection within the generation prompt. Furthermore, LATS lacks specialized prompts and strategies for fixing buggy programs.

---

[3]Due to the timing of our experiments, which were performed in April and May 2024, we replicate the results from the first version of the WorldCoder paper, which can be found at https://arxiv.org/abs/2402.12275v1. The authors have since developed a slightly different algorithm for code generation, which was published after we finalized our experiments. The original code generation algorithm based on Thompson Sampling, which we call WorldCoder in this work, was later published in Tang et al. [2024a].

Previous research has also focused on pseudocode-based reasoning, such as Parsel [Zelikman et al., 2023], which uses a custom pseudocode language to decompose the program into independent problems that can be solved separately. In contrast, we focus on the sequential refinement of solutions using a variant of MCTS and the environment's feedback to produce directly executable Python code that can be leveraged in model-based RL.

We refer the reader to Appendix G for further discussion on works that build language-conditioned world models but do not use code and on works that use programs as policies in RL.

## 3 Code World Models

In this Section, we first introduce the Code World Models framework and then the proposed Code World Models Benchmark.

**Code World Models framework.** Following the model-based Reinforcement Learning problem setting, we consider an environment represented by a Markov Decision Process with state space $\mathcal{S}$, action space $\mathcal{A}$, a transition function $p(s'|a, s)$, and a scalar reward function $R(s, a, s')$, with $s, s' \in \mathcal{S}$ indicating respectively the current and next state, and $a \in \mathcal{A}$ being the action taken from the current state. The task of a world model is to accurately represent $p$ and $R$. We make the following assumptions: 1) the environments are deterministic and fully observable, and 2) we are provided with a natural language description of the environment, which is detailed enough to infer the observation space as well as the logic of the transition and reward functions.

The first assumption implies a deterministic transition function $s' = f(s, a)$, rather than a probabilistic one as in the general case; we address this limitation in Section 6.1. The second assumption is akin to the situation where a human would be provided with an explanation, or a tutorial, about a task that they need to solve, in order to facilitate the learning process. Crucially, in a model-based scenario, we only need explanations about how the environment works, rather than requiring instructions about what to do in order to solve the task. Furthermore, we place ourselves in an offline RL scenario [Levine et al., 2020], assuming that a dataset $\mathcal{D}$ of $n$ one time-step transitions $\{(s, a, r, s', d)_i\}_{i=1,\ldots,n}$, where $d$ stands for the episode termination or *done* signal, is available, collected with some behavioural policy $\pi_B(a|s)$ in the environment of interest. However, this last assumption could be lifted, by using the Code World Model with a suitable planning algorithm to collect more trajectories from the environment, turning the algorithm into online RL, as done in Tang et al. [2024b].

**Code World Models Benchmark.** To comprehensively test world model generation for a variety of environments, we define a novel benchmark consisting of 18 RL environments of varying difficulty. We focus on commonly used environments of particular relevance to the RL community: classical control, physics-based PyGame environments and MuJoCo tasks. The environments' Python implementations as well as their documentation are adapted from the Gymnasium library [Towers et al., 2024]. The environments included in the resulting Code World Models Benchmark (CWMB) feature a mix of continuous and discrete action and observation spaces (more details in Appendix I).

For each environment, we collect a training dataset $\mathcal{D}$ of past trajectories. We curate $\mathcal{D}$ so that it includes at least some low-scoring and some relatively high-scoring behavior. However, we neither attempt to maximally cover the state space nor do we require optimal demonstrations. We aim to show that relatively low annotation effort is required to build CWMs: for the majority of environments, we collect just 5 trajectories equivalent to taking random actions and a further 5 suboptimal demonstrations exceeding some return threshold. As part of the benchmark, each transition $(s, a, r, s', d)$ in each resulting trajectory is used as an input-output sample to validate the generated models. The benchmark further includes a language description of each environment, derived from the documentation written for Gymnasium's end users (an example is included in Appendix N.3). A further discussion on how the quality of the collected dataset affects the performance of our method can be found in Appendix F.

## 4 GIF-MCTS

In this Section, we first specify the format of the Code World Models that we consider in this work and how we evaluate their accuracy. We then present Generate, Improve and Fix with Monte Carlo Tree Search (GIF-MCTS), a novel approach to leverage LLMs for code generation via multiple

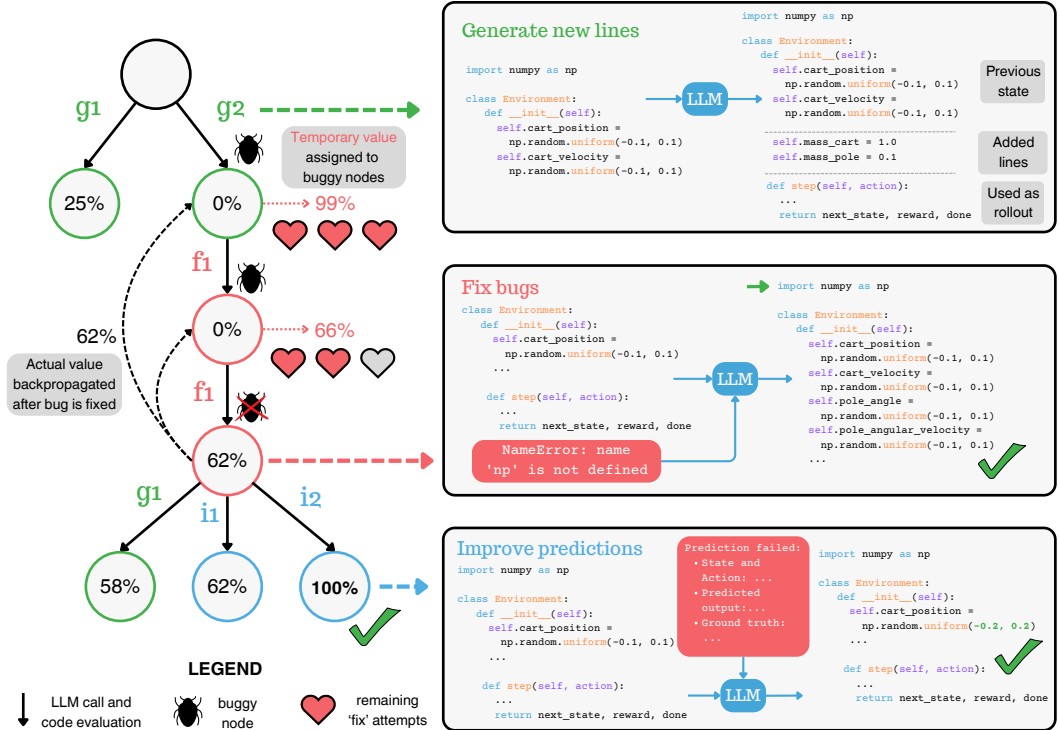

Figure 2: Example of a GIF-MCTS tree for generating a CWM. Starting from the root of the tree, every action taken corresponds to 1) prompting the LLM to either generate, improve or fix a CWM, 2) parsing the LLM completion, and 3) evaluating the CWM's correctness using the available environment trajectories as unit tests (presented as a percentage inside the nodes). On buggy nodes, we allow only fix actions for up to $f$ sequential attempts and replace the actual value with a temporary one, represented in red. In healthy nodes we allow only generate and improve actions. All action prompts are exemplified on the right. The number of total fix $f$ attempts is a model hyperparameter, set to three in this Figure and for our method.

sequential attempts in the presence of feedback, specifically tailored to the needs of building Code World Models.

We formulate the task of synthesizing a Code World Model as that of writing a Python `Environment` class with a `step()` function that jointly implements the transition and reward functions:

$$(\hat{s}', \hat{r}, \hat{d}) = \texttt{code\_environment.step}(s, a), \tag{1}$$

and consider a Code World Model correctly synthesized if it correctly reproduces all transitions in $\mathcal{D}$. We additionally define the accuracy $A$ of the Code World Model as the fraction of correctly predicted transitions (weighted uniformly on next state, reward and done signals) from the training dataset $\mathcal{D}$, or in other words:

$$A = \frac{1}{N} \sum_{i=1}^{N} \left( \frac{1}{3}\mathbf{1}[s_i', \hat{s}_i'] + \frac{1}{3}\mathbf{1}[r_i, \hat{r}_i] + \frac{1}{3}\mathbf{1}[d_i, \hat{d}_i] \right), \tag{2}$$

where $\mathbf{1}$ is the indicator function (equals to one if the pair is matching, zero otherwise) and $\hat{s}_i'$, $\hat{r}_i$ and $\hat{d}_i$ are the model's predictions.

GIF-MCTS takes as input the description of an environment, an LLM, environment trajectories and builds a tree to construct the code for the environment. Nodes in the tree are programs and edges are actions. Each action taken from a parent node produces a new complete program, which is split into a *state* part and a *rollout* part and stored in a child node. The child node's *state* is formed from the parent's state by appending $L$ additional lines of code (we set $L = 2$ in our work), while the rollout is the remaining part of the program, and represents one possible completion of the state, needed to evaluate (i.e., run) the code. This is a novel formulation of the state of a node, as we store

in the states partial programs in blocks of multiple lines, whereas previous work either stores only full programs [Zhou et al., 2023], or single tokens [Zhang et al., 2023]. The state represents the main flow of information from parent to child, while the rollout is used to estimate the expected accuracy of the child's state.

As in the standard MCTS algorithm, we perform multiple sequential iterations consisting of the following phases: selection, expansion, evaluation and value backpropagation. During the selection phase, starting from the root node, we use the Upper Confidence Bound for Trees (UCT) formula [Kocsis and Szepesvári, 2006] to select which action to take. If the corresponding node has never been expanded, we enter the expansion phase, otherwise we continue to apply the UCT formula to the actions of the new node. At expansion phase, we call the LLM to produce a program according to the type of action selected, parse the resulting program into the state and the rollout parts, and store both in the newly expanded node. We then compute the accuracy, defined above, using the rollout (evaluation phase), store the resulting value in the node, and backpropagate it to its ancestors. An example of a GIF-MCTS tree and the corresponding actions can be found in Figure 2.

With GIF-MCTS, we make the following contributions: 1) we present a novel framing of MCTS nodes and actions for long-form code generation in the presence of unit tests, 2) we propose three action types, specialised for code, whose added value we demonstrate through an ablation study, and 3) we propose a heuristic that empirically improves the trade-off between exploration and exploitation in the UCT formula used for action selection, balancing both explored and unexplored actions, and different action types (Appendix B). All these factors make GIF-MCTS specifically suitable for generating world models. Next we present the three action types (generate new lines, improve predictions and fix bugs) used in GIF-MCTS. We point the reader to the Appendix for the full action prompts, the remaining implementation details, and for the ablation study on the importance of the three action types.

### 4.1 GIF-MCTS Actions

**Generate new lines.** The goal of the *generate* action is to leverage the stochastic sampling ability of the LLM by generating varying continuations for a single code snippet in different branches of the tree, to fully explore the underlying space of possible solutions. The action prompt asks the LLM to generate the full code required to solve the task starting from the code stored in the node's *state*.

**Improve predictions.** Generating code in sequential blocks of lines can be too rigid if subtle or interdependent changes need to be made to the full program in order to pass more test cases and increase the reward. With the *improve* action, the LLM is prompted with the full program (*state* plus *rollout*) from the parent node, as well as one input example where the code did not behave as intended, along with the expected output. In the case of a Code World Model, this can be a wrongly predicted transition, with the input state and action taken by the agent, the ground-truth next state, and the model's predicted next state. The *improve* prompt also asks the LLM to produce a Chain-of-Thought explanation about where the current code is failing, and to attempt to fix the logic. The inclusion of both *generate* and *improve* actions allows GIF-MCTS to combine the advantages of block-wise incremental generation with the flexibility to backtrack and edit the whole program if needed.

**Fix bugs.** The code obtained with a *generate* or *improve* action will sometimes not be able to execute due to a syntax or runtime error, and will thus receive a reward of 0, strongly discouraging further exploration of the node. This can be wasteful, as sometimes the newly generated program can have sound logic and would receive a good reward if its bug(s) were removed. The *fix* action is tasked with resolving these bugs: the model is given the full program from the parent that encountered a bug along with feedback about the error and is asked to produce a fixed version of the code, aided by a Chain-of-Thought reasoning structure. To ensure that buggy nodes are chosen by the UCT formula, we assign them with temporary value until either the bug is fixed or no more attempts are allowed (see Appendix B for additional details).

## 5 Experiments

In this Section, we first describe the baseline code generation methods we compare against and then present empirical results on the APPS benchmark, the proposed CWMB and perform an additional

study on the RTFM environment. Additional ablations and qualitative results on GIF-MCTS are presented in Appendices C and D.

## 5.1 Baselines

The first baseline, denoted as **Zero-shot CoT** and used only for the experiments on APPS, adapts the work by Kojima et al. [2022] to code generation by appending *"Let's think step by step."* to the prompt and then parsing out from the completion only the code part. To report pass@20, we generate 20 independent completions for each problem, submit each of them, and count a problem as completed if at least one solution is correct.

The second baseline adapts the work by Tang et al. [2024b] to make as fair a comparison as possible. The **WorldCoder** algorithm calls the LLM with our *generate* prompt to produce an initial program, then for each remaining iteration we 1) select one of the previous programs as explained below, 2) refine it by calling the LLM with our *fix* prompt if the code has a bug, or our *improve* prompt otherwise, and 3) evaluate the resulting program against the unit tests. Each program $\rho$ is associated with a Beta distribution $B(\alpha, \beta)$ with initial parameters $\alpha = 1 + C * r(\rho)$ and $\beta = 1 + C(1 - r(\rho))$, which are updated every time the program is selected. Here $r(\rho)$ stands for the fraction of unit tests passed (same metric used in the evaluation phase of GIF-MCTS) and $C$ is a constant set to 5, as in the original work. To select the next program to be refined, one sample is drawn from each Beta distribution and the program with the highest score is selected. In all experiments, we use the same amount of calls of GIF-MCTS.

## 5.2 APPS

We assess the overall performance of GIF-MCTS for generic code synthesis in the presence of public unit tests on the APPS benchmark [Hendrycks et al., 2021], which consists of 10,000 Python coding problems in three categories of increasing difficulty: *Introductory*, *Interview* and *Competition*. We focus our evaluation on the hardest, *Competition* level test set comprised of 1000 problems, as it most closely reflects the challenges found in synthesizing CWMs: the problems tend to have a longer description, follow a specific format for the input and output, and include challenging logic. Early experiments on HumanEval [Chen et al., 2021], another popular coding benchmark, did not show a clear correlation between a model's performance on the benchmark and its ability to generate CWMs, as HumanEval problems are typically easier and solvable with much shorter code snippets.

As GIF-MCTS requires a reward signal from the environment, we make use of the suite of unit tests provided by APPS to evaluate the accuracy of a generated program. However, we note that the ground truth result from these tests is provided to GIF-MCTS with the *improve* action, and as such the model could simply memorize all possible results and return them without actually solving the problem. To avoid this, while we use all unit tests for computing the reward function, we only use samples from the first half as input-output examples for the *improve* action. In general, we use at least a fraction of the provided unit tests to evaluate every program generated during the GIF-MCTS loop, so our approach is only eligible for the pass@B metric, where B is the budget for the number of LLM calls used during the synthesis process. We leave extending the approach for pass@1 eligibility using self-generated unit tests [Chen et al., 2023] for future work. We report the strict accuracy rate (the fraction of problems on which all test cases are solved) on APPS for GIF-MCTS and other baselines in Table 1.

Table 1: **APPS competition results: comparison of methods.** We report the percentage of problems with all unit tests passed (*Strict Accuracy*). For our experiments, we also include the error of the mean on the percentage.

| Method | Model | Size | Strict Accuracy (%) | Evaluation Strategy |
|---|---|---|---|---|
| CodeRL [Le et al., 2022] | CodeT5 | 770M | 17.90 | pass@1000 |
| Parsel [Zelikman et al., 2023] | code-davinci-002 | N/A | 25.50 | pass@any |
| Zero-shot CoT * [Kojima et al., 2022] | Llama 3 | 70B | 23.2±1.3 | pass@20 |
| WorldCoder * [Tang et al., 2024b] | Llama 3 | 70B | 25.1±1.4 | pass@20 |
| GIF-MCTS (ours) | Llama 3 | 70B | **28.3±1.4** | pass@20 |

\* Our re-implementation.

**Results.** GIF-MCTS outperforms strong previous baselines on the APPS competition split, reaching a new state of the art to the best of our knowledge. While part of this can be due to advances in the underlying model, the comparisons with Zero-shot CoT and WorldCoder show improved performance over either prior method. GIF-MCTS is also markedly more sample efficient compared to established baselines; Parsel achieves the second best accuracy, but evaluates an exponentially growing number of solutions[4], while GIF-MCTS outperforms it by evaluating only 20 different programs.

## 5.3 Code World Models Benchmark

We evaluate our proposed GIF-MCTS approach and the WorldCoder baseline on the CWMB (introduced in Section 3). In this setting, we are interested in both the accuracy of the generated CWM, as well as its performance when actually employed by a planning algorithm. We use as accuracy the same metric used in the evaluation phase of GIF-MCTS (Section 4). To measure the performance of planning with the CWM, we define the normalized return $\mathcal{R}$ of a CWM as:

$$\mathcal{R}(\text{CWM}) = \frac{R(\pi_{\text{CWM}}) - R(\pi_{\text{rand}})}{R(\pi_{\text{true}}) - R(\pi_{\text{rand}})}, \tag{3}$$

where $R(\pi_{\text{CWM}})$ represents the return obtained when using the CWM as the internal model for the planner, $R(\pi_{\text{true}})$ is the return gathered with the true environment as the model while using the same planner (oracle planner), and $R(\pi_{\text{rand}})$ is the return from a random policy. This metric is positive when the performance of the CWM planner is above that of a random policy and reaches one when the return approaches the value from the oracle planner. We report results for the CWMB in Table 2. As the planner, we use a vanilla MCTS implementation for the environments with discrete actions and a Cross Entropy Method (CEM) planner [Rubinstein, 1997] for the ones with continuous action spaces (full details of the two planning algorithms are reported in Appendix L).

Table 2: **CWMB: main results.** For each method, we report the CWM accuracy and the normalized return $\mathcal{R}$, averaged separately across environments with discrete and continuous action spaces. Budget indicates the number of LLM calls. For each metric, we report the mean value across environments (and for the return, also across 10 episodes) with its error. For Llama 3, we report an average of three different random seeds for additional statistical significance.

| Model | Method | Budget | Discrete Action Space | | Continuous Action Space | |
|---|---|---|---|---|---|---|
| | | | Accuracy (↑) | $\mathcal{R}$(↑) | Accuracy (↑) | $\mathcal{R}$(↑) |
| Llama 3 70B (3 seeds) | GIF-MCTS (ours) | 50 | **0.84±0.03** | **0.76±0.03** | **0.35±0.03** | **0.22±0.01** |
| | WorldCoder * | 50 | 0.79±0.04 | 0.60±0.04 | 0.32±0.03 | 0.19±0.01 |
| GPT-4 Turbo (1 seed) | GIF-MCTS (ours) | 10 | **0.91±0.08** | **0.81±0.06** | **0.40±0.03** | **0.26±0.01** |
| | WorldCoder * | 10 | 0.87±0.09 | 0.79±0.06 | 0.24±0.06 | 0.20±0.01 |

\* Our re-implementation of [Tang et al., 2024b].

**Results.** Overall, GIF-MCTS outperforms WorldCoder for all environment splits and backbone models. For Llama 3, the most significant gains are made on the environments with discrete actions, while for GPT-4 on those with continuous actions. We speculate that, on discrete environments, Llama 3 makes better use of the budget with GIF-MCTS than with WorldCoder, whereas GPT-4 saturates its performance in both cases. On the other hand, on the harder environments with continuous actions, Llama 3 hits a performance ceiling in both cases, while GPT-4 leads to higher improvements with our method. For example, Llama 3 was unable to generate a fully executable CWM (with either method) for the two hardest environments, Humanoid-v4 and HumanoidStandup-v4, due to their complexity and large observation space, while GPT-4 successfully generated a model for each environment in the benchmark.

## 5.4 Read to Fight Monsters

We perform an additional experiment on the Read to Fight Monsters (RTFM) grid-world environment, first introduced by Zhong et al. [2020] for testing grounded language understanding in RL. Every

---

[4]Results reported for Parsel use 8 pseudo-codes per problem, each implementing $n$ sub-functions (with $n$ being problem-dependent) 16 times and then evaluating up to $8 * 16^n$ sub-functions combinations against APPS unit tests and keeping the best result.

episode presents two monsters belonging to two teams, and two items, each effective on a specific monster. The environment provides the agent with a written descriptions of the task dynamics (also called manual), describing monsters' weaknesses and membership to teams, and a goal (which team of monsters to defeat). Crucially, the agent needs to perform multi-step reasoning between such information and the current state of the environment to figure out a plan of action (for more details we refer to the original work by Zhong et al. [2020]). We consider a version of the environment where we fix the input manual, meaning all relationships between items and monsters are fixed across episodes, and we don't allow the monsters to move, as their patterns are stochastic. This isolates the natural language understanding component of the task, while we leave to future work to demonstrate the applicability of the CWM framework to the full RTFM task.

We report the results on the simplified RTFM environment in Table 3, using MCTS as a planner for computing the normalized returns. We further experiment with a higher number of LLM calls for GPT-4 Turbo, matching the one used for Llama 3, as we couldn't do this on the full CWMB due to budget concerns.

Table 3: **RTFM results.** For each method and computational budget (LLM calls), we report the CWM accuracy and the normalized return $\mathcal{R}$ (computed across 10 episodes), with their errors.

| Model | Method | Budget | Accuracy ($\uparrow$) | $\mathcal{R}(\uparrow)$ |
|---|---|---|---|---|
| Llama 3 70B | GIF-MCTS (ours) | 50 | **0.58 $\pm$ 0.02** | **-0.11 $\pm$ 0.12** |
| | WorldCoder * | 50 | 0.23 $\pm$ 0.01 | **-0.11 $\pm$ 0.12** |
| GPT-4 Turbo | GIF-MCTS (ours) | 10 | **0.71 $\pm$ 0.01** | **0.31 $\pm$ 0.19** |
| | WorldCoder * | 10 | 0.33 $\pm$ 0.01 | 0.22 $\pm$ 0.18 |
| GPT-4 Turbo | GIF-MCTS (ours) | 50 | **1.00 $\pm$ 0.00** | **1.00 $\pm$ 0.00** |
| | WorldCoder * | 50 | 0.64 $\pm$ 0.02 | -0.06 $\pm$ 0.12 |

\* Our re-implementation of [Tang et al., 2024b].

**Results.** GIF-MCTS outperforms WorldCoder under all settings by a significant margin in terms of accuracy, but the generated CWM is only able to match the performance of the ground-truth simulator when the program is perfect. This highlights the necessity of completely accurate predictions, as further discussed in Section 6, while also providing empirical validation for the scaling properties of the approach: as GIF-MCTS is allowed more calls, it manages to refine the CWM it generated with a lower budget. As this version of the RTFM environment has never been published, this experiment can also alleviate concerns that the final CWM was memorized by the LLM during pre-training. We present and discuss further evidence against the significance of data contamination in Appendix E.

## 6 Discussion

In this section, we first discuss some takeaways from the empirical results and then elaborate on some of the limitations for our method.

**GIF-MCTS vs. WorldCoder.** We believe that GIF-MCTS outperforms WorldCoder because it produces a more diverse set of programs. WorldCoder initially generates a single program from scratch and then samples and refines a complete program in each iteration. In contrast, GIF-MCTS can generate multiple programs either from scratch or from partial programs by taking the *generate new lines* action at the root node or subsequent nodes. This approach better explores the solution space, leading to improved performance. Our ablation study *No Generate action* in Table 6 of the Appendix supports this finding. This study uses a tree search like GIF-MCTS but always refines a complete program, similar to WorldCoder, and results in lower performance compared to our method.

**Accuracy-Return Gap.** We observe empirically from Table 2 that the CWM accuracy is always higher than its normalized return, and the two metrics match only when the CWM is flawless. This is often due to the incorrect prediction of terminal states: these are rarer in the replay buffer, especially states that terminate with a success/positive reward. This can cause the planning algorithm to fail, as it is missing the reward signal. Part of the performance gap could also be due to sparse coverage of the environment by the collected trajectories. Individual results for each environment elaborating

on this are included in Appendix J. Future work could explore retrieving and combining different CWMs that complement each other to improve the performance on important edge cases.

**Sample Efficiency.** Generating a CWM requires far less interaction with the environment than traditional model-based approaches. As the gathered transitions are only used to validate the program and as in-context examples, a small curated set (enough to cover possible edge cases and different reward values) is enough to properly validate the generated code. In our experiments we only gather 10 trajectories made up of at most 100 steps as the offline dataset, while benchmarks specifically designed to challenge for sample efficiency [Bellemare et al., 2013] require agents to use at most 100k frames, which is two orders of magnitude higher. We leave more thorough experiments on sample efficiency for CWM agents to future work.

**Comparison with Offline RL.** We expect CWMs to hold advantages over classical RL methods in regimes with scarce data and environments that can be easily described by language and modeled with code. We report in Appendix K a preliminary comparison on the CWMB of the return achieved with our CWMs or with a SOTA offline RL method, Conservative Q-Learning (CQL) [Kumar et al., 2020], trained on the same amount of trajectories used for synthesizing the CWMs. We find that CWMs compare favourably against CQL on environments with discrete action spaces, while CQL's performance is superior on the continuous action space environments, which are harder to model. RL methods, including CQL, would likely benefit from more experience, as they overfit with scarce data.

## 6.1 Limitations

**Code World Models.** The CWMs framework is an exciting direction for model-based planning, but we still rely on limiting assumptions of deterministic and fully observable environments. Both stochasticity and partial observability would pose challenges, especially on the verification of the CWM prediction, as there is no set result for a given input. We leave extending the approach to account for both stochastic and partially observable environments to future work.

Another potential issue is providing a description of the environment that can be reasonably converted to a Python function (e.g. a manual documenting key variables) when such a description is not available (e.g. when the environment is defined with image observations). Previous work has begun to tackle this issue [Migimatsu and Bohg, 2022] and preprocessing techniques such as image-to-text models [Ren et al., 2024] could be used to address this problem in future work.

Code-based models may also be too rigid when the environment requires adapting to changing dynamics, which would imply rewriting the CWM on the fly. A possible solution could be breaking down the CWM into smaller functions that can be re-written individually by an LLM, to account for some changes in the environment, or modeling variable factors as arguments to the step function. CWMs struggle especially on complex physics-based environments; thus a promising direction could also be allowing programs generated by GIF-MCTS to make use of external tools and libraries, such as physics simulators.

**GIF-MCTS.** We have validated the GIF-MCTS approach as an efficient code synthesis method, with the key limiting assumption of having available test cases to evaluate code, which could be difficult to provide in certain tasks. In those cases, it would be possible to use self-generated test cases [Chen et al., 2023], but since this does not reflect the CWM setting we leave this for future work.

## 7 Conclusion

We present Code World Models, a general framework to leverage LLMs to build world models for RL agents. We further show that GIF-MCTS is a strong code synthesis method, able to successfully integrate external feedback to self-debug and improve code, demonstrating examples of world modeling and downstream planning for a range of environments. We are confident that the Code World Models approach will lead to the development of fast, interpretable and sample efficient model-based RL agents, exploiting the strengths provided by increasingly powerful LLMs, without directly predicting the environment dynamics with them. We are hopeful that improvements to both the underlying LLM backbone and refinements to the code generation method itself will result in powerful Code World Models for even more complex environments than those treated in this work.

## Acknowledgments and Disclosure of Funding

This work was supported by the Research Council of Finland (Flagship programme: Finnish Center for Artificial Intelligence FCAI, and grants 352986, 358246) and EU (H2020 grant 101016775 and NextGenerationEU). We acknowledge CSC for awarding this project access to the LUMI supercomputer, owned by the EuroHPC Joint Undertaking, hosted by CSC (Finland) and the LUMI consortium through Finland.

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

## A    Broader Impact

The CWM framework enables LLMs to generate world models for model-based Reinforcement Learning, which could potentially be employed for planning with a real agent. As the code generated by the LLM is untrusted, it should always be checked by a human expert before it is used under any circumstances. Alternatively, as CWMs are represented with Python code, this also allows for interpretable world models, which could be safer for critical applications after being vetted by an expert.

## B    Additional GIF-MCTS implementation details

**Choice of Actions**    If a node doesn't contain a bug, new *generate* and *improve* actions should always be available (with the exception of the root node, which will only have a new *generate* action, since there is no pre-existing code to improve). After an action is expanded, we add a new action of the same type to the parent node, so that the tree can have a variable number of nodes at any level. By contrast, a buggy node will only ever have a single *fix* action available, and no new *fix* actions will be added to the parent, enforcing the fixes to be applied sequentially (as there is no need to expand the tree horizontally in a buggy node). To select actions, we follow a modified variant of the Upper Confidence Bound for Trees (UCT) formula [Kocsis and Szepesvári, 2006] as follows:

$$\text{UCT}(\text{node}_i) = v_i + C \cdot \sqrt{\frac{\ln N_i}{n_{a=a_i} + \epsilon}},$$

where $v_i$ is the value of the node, $C$ is a constant parameter used to balance exploration (empirically set to 0.1), $N_i$ is the number of visits to the node's parent and $n_{a=a_i}$ is the number of expanded children with the same action type (relative to the parent). This last parameter is required to avoid trees that only grow horizontally due to the added actions: if a single action is chosen too many times from the same parent, the $n_{a=a_i}$ term will cause the exploration value for new nodes for the same action to keep decreasing and therefore encourage more exploration.

**Value Estimation for Unexplored Nodes.**    Nodes that have not yet been visited are missing their value, which prevents the application of the UCT formula. To circumvent this, we employ a simple linear model, trained during the overall search, to predict the value of unexplored nodes. This estimate is specific to an action type, so that each has a separate classifier, and further differentiates local and global values. We define the global value $v_G$ as the average of all values of the nodes with the same action type at any level of the tree and the local value $v_L$ as the average of all expanded children with the same action type. The linear model then simply learns to predict the value $v_i$ of a given action as a balanced sum of the two values, normalized between zero and one, with the following formula:

$$v_i = \frac{w_G \cdot v_G + w_L \cdot v_L}{w_G + w_L},$$

where the $w_G$ and $w_L$ parameters are learned during the search using gradient descent.

Initially, the global average $v_G$ will also be empty, which would cause the first values to be ill-defined. To mitigate this, we initialize the global average with a prior value which we tune empirically. To ensure a single unlucky generation does not prematurely downweight an action type, this prior is further assigned an initial count, used to weight the prior when computing the average (effectively acting as if there were $n$ nodes already discovered with the prior value).

**Value Estimation for Buggy Nodes.**    As mentioned in Sec. 4, buggy nodes will get a reward of 0 and would thus never be explored. To allow the *fix* action to be chosen, we assign a temporary value to the buggy node (which is effectively the parent of the fix action nodes). This can be chosen arbitrarily to trade-off attempting to fix buggy nodes (exploration) and focusing on other already functioning branches (exploitation). In our implementation, we initially set this value to 0.99, effectively forcing the model to attempt fixing a buggy node at least once. Naturally, a program can have more than one bug which could require the method taking multiple *fix* actions. To account for this, if the outcome of a *fix* action is still a bug, we gradually linearly decrease the temporary value of the parent until it reaches zero after a certain number of allowed fixes $f$, which we set to three. After $f$ unsuccessful fixes, the temporary value is set to zero, which strongly discourages the buggy parent node from

being selected again. Otherwise, the value of the buggy parent and the *fix* children are set to the value received by the newly fixed program. It is also important to note that the temporary values are excluded from the backtracking step of the MCTS algorithm, to avoid skewing the ancestors' values.

**Hyperparameters**   We report all hyperparameters used for GIF-MCTS as well as their description in Table 4, while hyperparameters related to the backbone LLM are reported in Table 5. We refer to the Huggingface documentation[5] for an accurate description of each LLM parameter.

Table 4: **GIF-MCTS hyperparameters.**

| Parameter | Description | Value |
|---|---|---|
| $L$ | Number of new lines extracted from a generate action. | 2 |
| $\epsilon$ | Visit count offset. | 1.0 |
| $C$ | Exploration constant. | 0.1 |
| $\gamma$ | Discount factor. | 1.0 |
| $v_g$ | Initial prior for generate actions (with its initial count). | 0.5 (2) |
| $v_i$ | Initial prior for improve actions (with its initial count). | 0.55 (2) |
| $f$ | Number of allowed fixes to a node. | 3 |

Table 5: **Llama 3 hyperparameters.** Note that for GPT-4 Turbo, the only parameter used was the number of maximum new tokens, set to the same value used for Llama.

| Parameter | Value |
|---|---|
| max_new_tokens | 1500 |
| temperature | 1.0 |
| top_k | 100 |
| top_p | 0.8 |
| num_return_sequences | 1 |
| num_beams | 1 |

## C   Ablation Study on GIF-MCTS

We perform an ablation study to validate the individual contribution of each action type of GIF-MCTS. We run the MCTS procedure on CWMB with only two out of the three actions available and compare the accuracy with the full method in Table 6. Note that for the Fix and Improve MCTS variant, one *generate* action is applied at the root node to obtain an initial program, which the algorithm expands from with the available budget. All ablations are performed using Llama 3 70B. For budget constraints, we run a single random seed for each ablation and compare with a single GIF-MCTS run with the same random seed.

**Results.**   The performance of the method drops after the removal of each action, most significantly in the harder set of continuous environments (while there is more statistical uncertainty for the discrete environments). Fixing bugs appears to be the most important action: it is much more efficient to try fixing a bug aided by external feedback compared to blindly generating the same code snippet until bug-free. As the complexity of the environment grows, it might also become increasingly challenging to generate a fully functioning program from the start. On the other hand, *improve* seems to be the least impactful: this makes sense, as intuitively improving a code snippet that already works is has less room for improvement.

## D   Qualitative Study

To investigate the specific effectiveness of each individual type of action, we analyze the trees produced by GIF-MCTS and report some statistics of interest in Table 7. We specifically focus on the

---

[5]https://huggingface.co/docs/transformers/main_classes/text_generation

Table 6: **CWMB results: ablation study.** We compare the full GIF-MCTS method against three ablated variants, each leaving out one of the three action types. For each method, we report the CWM accuracy and the normalized return $\mathcal{R}$, averaged separately across environments with discrete and continuous action spaces. For each metric we report the mean value across environments (and for the return, also across 10 episodes) with its error.

| Method | Budget | Discrete Action Space | | Continuous Action Space | |
|---|---|---|---|---|---|
| | | Accuracy ($\uparrow$) | $\mathcal{R}(\uparrow)$ | Accuracy ($\uparrow$) | $\mathcal{R}(\uparrow)$ |
| GIF-MCTS (ours) | 50 | **0.88±0.07** | **0.83±0.06** | **0.38±0.04** | **0.23±0.02** |
| No *Generate* action | 50 | 0.87±0.07 | 0.73±0.09 | 0.25±0.06 | 0.16±0.01 |
| No *Improve* action | 50 | 0.85±0.06 | 0.79±0.07 | 0.34±0.05 | 0.17±0.02 |
| No *Fix* action | 50 | 0.81±0.08 | 0.55±0.05 | 0.21±0.08 | 0.10±0.01 |

difference in the overall distribution of action types in the tree as a whole compared to the actions chosen on the path that led to the best result, which can be used to find specific biases towards a specific action.

Table 7: **Qualitative Analysis.** We report a qualitative study for the frequency with which GIF-MCTS chooses each type of action on average. The first section of the table is considering the whole tree, while the second section (*path* quantities) only consider the path from the root node to the node with the highest value (where the code used as the environment was generated).

| Quantity | Discrete Action Space | | Continuous Action Space | |
|---|---|---|---|---|
| | Llama 3 70B | GPT-4 Turbo | Llama 3 70B | GPT-4 Turbo |
| % generates | 50.0 | 88.3 | 18.5 | 33.4 |
| % improves | 44.7 | 8.3 | 35.3 | 34.8 |
| % fixes | 5.3 | 3.4 | 46.2 | 31.8 |
| Path length | 5.7 | 2.3 | 3.2 | 2.3 |
| % path generates | 73.2 | 100.0 | 47.0 | 59.0 |
| % path improves | 17.5 | 0.0 | 5.0 | 6.3 |
| % path fixes | 9.3 | 0.0 | 48.0 | 34.7 |
| Tree depth | 15.6 | 5.0 | 10.8 | 4.5 |

From the results, the method presents a pretty clear bias towards the *generate* action at the expense of the *improve* action on the optimal path. While the model tries to improve its previous code reasonably often (more than 35% of the times in most cases) the percentage of these actions that actually led to the best node drops significantly in the optimal path, which could imply that generate actions are the most effective.

With a closer inspection into the trees themselves, we find that often there is an initial set of *generate* actions that already result in values that are close to the maximum found by the tree, and then later *improve* actions are chosen thanks to the same-action penalty term in the modified UCT formula, which can result in marginal increases (as they are only refining code that is already promising) or fail to improve the previous program (as the logic might be hard to extrapolate). As such, many *improve* actions are needed in order to find a sample that is actually increasing the performance, while *generate* actions have the advantage of being chosen at the beginning, where it is possibly easier to find good programs.

Still, the fact that many *improve* actions are taken that result in either the same value as the previous node or at times even in worse accuracy is a potential bottleneck for the method, which seems to corroborate recent evidence [Olausson et al., 2023] showing that LLMs are often unable to provide proper feedback on their own code generations. Stronger models might thus be needed to specifically analyze and criticize the code (e.g. one model specialized in explaining code which provides feedback to another one specialized in generating it).

There is also a clear difference between the set of easier discrete action space problems, for which the percentage of *fix* actions is very low (with GPT-4 Turbo only needing generates in order to synthesize perfect or near-perfect models, as shown in Table 11) and the harder continuous action space problems, where fixing bugs becomes much more prominent.

## E Data Contamination

With any experiment involving LLMs there is a concern about data contamination: the model's pre-training corpus could have included the original implementation for the various programs we are trying to generate, which means that hypothetically the model could simply be memorizing them and repeating them. To alleviate these concerns, we analyze each experiment individually:

- For the APPS benchmark, the programming problems we used are sourced from three main websites. The benchmark authors managed to crawl reference solutions for only two of these sites (*AtCoder* and *Codeforces*, which include 264 and 41 problems respectively). This means that for the third website, *Kattis*, which makes up a majority of the benchmark with 691 problems, no reference solution can be found online (and thus likely also not in the training corpus for the LLMs).

  Performance across all methods and models in the competition split is correlated with the source websites of the problems, but not with the availability of the solutions: the highest results are obtained from Kattis (0.347 strict accuracy rate), the only site where solutions are not available online. Notably, all methods and models achieve a 0% pass rate for the 41 problems from AtCoder, for which reference solutions are available online. This suggests that the difficulty of the various sources is more important than the reference solution.

- While we observe that some parts of the generated CWMB environments recall implementations available online (e.g., constants' values in the CartPole environment), the logic of the step function remains distinct from the reference model. Furthermore, the MuJoCo-based environments used the simulator in the official implementation, which is not available in our setting, so the code is necessarily different. Examples of generated CWMs along with their ground-truth implementations can be found in Appendix O for a more thorough comparison.

- As we use a modified version of the RTFM environment (with fixed manuals and no stochasticity), there is no reference solution for it online, which provides evidence that our solution is not merely retrieving information from the LLM's training data.

Generally speaking, there is of course no way to outright dismiss these concerns. However, our method is compared to baselines using the same underlying models, ensuring that the superior performance reported for GIF-MCTS is not biased by potential data contamination.

## F Data Quality

As part of the CWMB, for each environment the collected dataset $\mathcal{D}$ contains both low-scoring and high-scoring trajectories. As discussed in Section 3, this is fairly standard practice for offline RL, as the general assumption is that in the real world large datasets can be collected from a very diverse ensemble of sources. While it would be expected that at least one example for all possible outcomes is required in order for the world model to be precise and comprehensive, our approach can in principle learn a fair model even in hard environments when provided with only a few random trajectories by leveraging the language description provided to the LLM when generating the program. This could theoretically be used to generalize the rules of the environment outside of the observed transitions: the model does not need to see what happens if it can read about it.

We performed an additional experiment on RTFM: we collected 10 trajectories all resulting in failures, so that a reward of +1 is never observed. In other words, this is a worse version of the same buffer used for the main experiment, which by construction carries less information. We synthesized a CWM with GIF-MCTS and GPT-4 using 50 calls, which in the original experiment resulted in a perfect model (Section 5.4). The resulting CWM is 100% accurate on the newly collected dataset and even correctly predicts a reward of +1 for positive transitions, which are not included in the dataset, thanks to the language description. When tested on the original dataset $\mathcal{D}$ from the CWMB (which

contains both positive and negative rewards), the model still scores 100% accuracy, on par with the model generated with the full range of data.

## G    Additional Related Work

We expand in the following the Related Work section, covering the works that try to build world models with language and those who explored using programs to express RL policies.

**World Models with Language.**   Model-based RL methods are built around learning a predictive model of the environment to inform the agent's decisions [Sutton, 1991]. A recently growing body of research is focusing on building world models that can include information in natural language, as opposed to approaches using only vision or full state observations [Hafner et al., 2021]. Dynalang [Lin et al., 2024] predicts the future text and image representation of the environment with an encoder-decoder architecture with a joint input of previous frames and text, while Zhang et al. [2024] formulate the modeling task as an autoregressive prediction task performed by a Transformer [Vaswani et al., 2017]. Voltron [Karamcheti et al., 2023] also uses an encoder-decoder model for language-driven representation learning for robotics. Other promising avenues include predicting the pixels in the next image observation [Yang et al., 2024, Bruce et al., 2024, Micheli et al., 2023, Liu et al., 2024].

**Programmatic RL.**   Verma et al. [2018, 2019] first introduced Programmatically Interpretable RL (PIRL), which focuses on representing RL policies as interpretable and verifiable programs by first learning an oracle policy with deep RL and then distilling a program with a domain specific language that can model tree-like programs. Similarly, Bastani et al. [2018] focus on extracting decision trees from an oracle policy with imitation learning and Inala et al. [2020] use finite-state automata, which can also include advanced control structures such as loops, with Silver et al. [2020] similarly using a language with a token that can perform loops. The need for an oracle was later removed by Qiu and Zhu [2022] by directly optimizing differentiable programs. Later, Trivedi et al. [2021] introduce LEAPS, which uses a Variational Auto-Encoder (VAE) to embed programs into a latent space and search new programs in the latent space, further extended by Liu et al. [2023] with the use of Hierarchical RL that composes simple programs together in order to generalize to out of distribution codes not seen by the VAE. However, Carvalho et al. [2024] has recently shown that the latent space is actually harder for optimization algorithms, and that simply performing the search in the program space leads to better results. Azad et al. [2022] instead proposed using a similar domain specific language to build a world model, with a similar approach presented by EMPA [Tsividis et al., 2021]. As these methods all use traditional program synthesis methods to generate their code, recent works have also looked into using LLMs to generate RL policies. Liang et al. [2023] uses Python code to interface with APIs and generate a robotic policy, with a similar approach concurrently introduced by Singh et al. [2023]. Voyager [Wang et al., 2023] generates an incrementally growing skill library using JavaScript code to play Minecraft.

## H    Comparison of Inference Times

We further demonstrate the efficiency of CWMs compared to directly using an LLM as the world model in Table 8. On a selection of three environments from the CWMB we ask GPT-4 Turbo to directly predict the next observation of the environment given its description and some in-context examples of the task, and compare the inference time with calling the step function of the CWM. Calling the Python program is four orders of magnitude quicker for the easiest environment and seven orders of magnitude quicker for the hardest environment. We additionally observe that none of the predictions made by GPT-4 Turbo were accurate.

## I    Code World Models Benchmark Details

We include a detailed list of statistics for each environment in the CWMB in Table 9. Notice that when creating the descriptions from the Gymnasium docstrings, we left out documentation sections that do not relate to the environment definition itself, such as versioning information, Gymnasium-related arguments, and external references, from these descriptions. For the reported number of tokens we

Table 8: **Comparison:** inference times between GPT-4 and CWM. Results are calculated from a sample of 10 transitions from the replay buffer used during GIF-MCTS.

| Environment | GPT-4 Time (s) | CWM Time (s) |
|---|---|---|
| CartPole-v1 | 2.2 | 0.00005 |
| HalfCheetah-v4 | 6.1 | 0.0001 |
| Humanoid-v4 | 146.7 | 0.0001 |

choose OpenAI's open source `tiktoken` tokenizer[6]. The code lines and code tokens are reported from the corresponding CWM generated by GPT-4 Turbo using GIF-MCTS with a budget of 10. This is meant to be a general indication of how long a typical implementation of the environment would be, but can of course vary. All environment descriptions were parsed from Gymnasium v.0.29.1.

Table 9: **CWMB details.** Detailed statistics for each environment in the CWMB. An Action Space or Observation Space indicated between bars ($|\mathcal{A}|$, $|\mathcal{S}| = n$) indicate a discrete space with $n$ different choices. The value intervals for each space are omitted for visual clarity.

| Environment | Description Lines | Description Tokens | Action Space Dimensionality | Observation Space Dimensionality | Code Lines* | Code Tokens* |
|---|---|---|---|---|---|---|
| Blackjack-v1 | 66 | 601 | $|\mathcal{A}| = 2$ | $|\mathcal{S}| = (32, 11, 2)$ | 94 | 826 |
| CliffWalking-v0 | 47 | 456 | $|\mathcal{A}| = 4$ | $|\mathcal{S}| = 48$ | 61 | 483 |
| Taxi-v3 | 89 | 724 | $|\mathcal{A}| = 6$ | $|\mathcal{S}| = 500$ | 83 | 767 |
| Acrobot-v1 | 66 | 859 | $|\mathcal{A}| = 3$ | $\mathcal{S} \in \mathbb{R}^6$ | 76 | 794 |
| CartPole-v1 | 53 | 663 | $|\mathcal{A}| = 2$ | $\mathcal{S} \in \mathbb{R}^4$ | 62 | 639 |
| MountainCar-v0 | 47 | 454 | $|\mathcal{A}| = 3$ | $\mathcal{S} \in \mathbb{R}^2$ | 62 | 426 |
| Ant-v4 | 148 | 2983 | $\mathcal{A} \in \mathbb{R}^8$ | $\mathcal{S} \in \mathbb{R}^{27}$ | 33 | 267 |
| HalfCheetah-v4 | 86 | 1674 | $\mathcal{A} \in \mathbb{R}^6$ | $\mathcal{S} \in \mathbb{R}^{17}$ | 58 | 554 |
| Hopper-v4 | 87 | 1529 | $\mathcal{A} \in \mathbb{R}^3$ | $\mathcal{S} \in \mathbb{R}^{11}$ | 91 | 847 |
| Humanoid-v4 | 204 | 4578 | $\mathcal{A} \in \mathbb{R}^{17}$ | $\mathcal{S} \in \mathbb{R}^{376}$ | 68 | 617 |
| HumanoidStandup-v4 | 202 | 4551 | $\mathcal{A} \in \mathbb{R}^{17}$ | $\mathcal{S} \in \mathbb{R}^{376}$ | 50 | 442 |
| InvertedDoublePendulum-v4 | 84 | 1364 | $\mathcal{A} \in \mathbb{R}^1$ | $\mathcal{S} \in \mathbb{R}^{11}$ | 54 | 465 |
| InvertedPendulum-v4 | 55 | 683 | $\mathcal{A} \in \mathbb{R}^1$ | $\mathcal{S} \in \mathbb{R}^4$ | 66 | 633 |
| Pendulum-v1 | 50 | 545 | $\mathcal{A} \in \mathbb{R}^1$ | $\mathcal{S} \in \mathbb{R}^3$ | 58 | 500 |
| Pusher-v4 | 98 | 2035 | $\mathcal{A} \in \mathbb{R}^7$ | $\mathcal{S} \in \mathbb{R}^{23}$ | 76 | 587 |
| Reacher-v4 | 87 | 1472 | $\mathcal{A} \in \mathbb{R}^2$ | $\mathcal{S} \in \mathbb{R}^{11}$ | 78 | 699 |
| Swimmer-v4 | 68 | 1168 | $\mathcal{A} \in \mathbb{R}^2$ | $\mathcal{S} \in \mathbb{R}^8$ | 80 | 700 |
| Walker2d-v4 | 92 | 1785 | $\mathcal{A} \in \mathbb{R}^6$ | $\mathcal{S} \in \mathbb{R}^{17}$ | 81 | 770 |

* Indicative number sampled from a single result, can vary.

## J    Results for Individual Environments

We report the individual accuracy and return for each environment in the CWM when using Llama 3 in Table 10 and when using GPT-4 Turbo in Table 11.

## K    Comparison with Offline RL

We compare the overall performance of a SOTA offline RL method, Conservative Q-Learning (CQL) [Kumar et al., 2020], against a planning agent using the synthesized CWM with our method. We report in Table 12 the average raw reward obtained over 10 episodes for a random policy, CQL, planning agents with the CWM obtained by GIF-MCTS (ours) respectively with Llama 3 and GPT-4, and a planning agent with oracle access to the true environment. CQL was trained with 10 epochs for 100 steps per epoch (1000 total) using the *same dataset* $\mathcal{D}$ used to learn our CWMs. We chose 1000 steps to match the data to gradient steps ratio from the original CQL paper. Since our replay buffers are much smaller (the original paper worked with D4RL [Fu et al., 2020], which provides 1M transitions per task), we started to observe severe overfitting for CQL with more training steps.

Overall, there is a balance between CQL and CWMs, with CWMs being more suited to discrete tasks and CQL outperforming CWMs in complex physics tasks, where our method struggles. However,

---

[6]https://pypi.org/project/tiktoken/

Table 10: **CWMB results.** Individual results for each environment in the CWMB using Llama 3 (we report the results for the first seed only).

| Environment | Action Space | GIF-MCTS | | WorldCoder | |
|---|---|---|---|---|---|
| | | Accuracy (↑) | $\mathcal{R}$(↑) | Accuracy (↑) | $\mathcal{R}$(↑) |
| CartPole-v1 | Discrete | 1.00 | 1.11 | 0.92 | 1.09 |
| CliffWalking-v0 | Discrete | 1.00 | 1.01 | 1.00 | 0.97 |
| MountainCar-v0 | Discrete | 1.00 | N/A | 0.83 | N/A |
| Taxi-v3 | Discrete | 0.92 | 0.67 | 0.44 | 0.23 |
| Blackjack-v1 | Discrete | 0.83 | 0.53 | 0.85 | 0.41 |
| Acrobot-v1 | Discrete | 0.54 | N/A | 0.73 | N/A |
| InvertedPendulum-v4 | Continuous | 0.66 | 0.14 | 0.66 | 0.01 |
| Pusher-v4 | Continuous | 0.41 | 0.74 | 0.41 | 0.77 |
| Pendulum-v1 | Continuous | 0.34 | -0.15 | 0.31 | -0.15 |
| Walker2d-v4 | Continuous | 0.34 | 0.07 | 0.34 | 0.08 |
| Hopper-v4 | Continuous | 0.33 | 0.15 | 0.00 | 0.02 |
| Swimmer-v4 | Continuous | 0.33 | 0.01 | 0.33 | 0.07 |
| HalfCheetah-v4 | Continuous | 0.33 | 0.13 | 0.33 | 0.15 |
| Ant-v4 | Continuous | 0.33 | 0.67 | 0.33 | 0.69 |
| InvertedDoublePendulum-v4 | Continuous | 0.25 | 0.06 | 0.34 | 0.05 |
| Reacher-v4 | Continuous | 0.13 | 0.93 | 0.42 | 0.67 |
| HumanoidStandup-v4 | Continuous | N/A | 0.00 | N/A | 0.00 |
| Humanoid-v4 | Continuous | N/A | 0.00 | N/A | 0.00 |

Table 11: **CWMB results.** Individual results for each environment in the CWMB using GPT-4 Turbo.

| Environment | Action Space | GIF-MCTS | | WorldCoder | |
|---|---|---|---|---|---|
| | | Accuracy (↑) | $\mathcal{R}$(↑) | Accuracy (↑) | $\mathcal{R}$(↑) |
| CartPole-v1 | Discrete | 1.00 | 0.99 | 1.00 | 1.00 |
| CliffWalking-v0 | Discrete | 1.00 | 0.98 | 1.00 | 0.89 |
| MountainCar-v0 | Discrete | 1.00 | N/A | 1.00 | N/A |
| Taxi-v3 | Discrete | 0.99 | 0.87 | 0.99 | 0.67 |
| Blackjack-v1 | Discrete | 0.93 | 0.41 | 0.79 | 0.59 |
| Acrobot-v1 | Discrete | 0.53 | N/A | 0.42 | N/A |
| InvertedPendulum-v4 | Continuous | 0.66 | 0.08 | 0.66 | 0.00 |
| Humanoid-v4 | Continuous | 0.43 | 0.01 | 0.00 | 0.00 |
| HumanoidStandup-v4 | Continuous | 0.42 | -0.04 | 0.00 | 0.00 |
| Reacher-v4 | Continuous | 0.42 | 0.88 | 0.42 | 0.71 |
| Pusher-v4 | Continuous | 0.41 | 0.72 | 0.41 | 0.70 |
| InvertedDoublePendulum-v4 | Continuous | 0.41 | 0.02 | 0.00 | 0.00 |
| Pendulum-v1 | Continuous | 0.38 | 0.51 | 0.38 | 0.50 |
| Walker2d-v4 | Continuous | 0.34 | 0.03 | 0.01 | 0.03 |
| Hopper-v4 | Continuous | 0.34 | -0.04 | 0.33 | -0.01 |
| Swimmer-v4 | Continuous | 0.33 | 0.04 | 0.33 | 0.02 |
| HalfCheetah-v4 | Continuous | 0.33 | 0.23. | 0.33 | 0.24 |
| Ant-v4 | Continuous | 0.33 | 0.69 | 0.00 | 0.20 |

CWMs also reach competitive results in some of these harder environments, such as `Pendulum-v1`, `Reacher-v4` and to a lesser extent `Ant-v4`, `Pusher-v4` and `HalfCheetah-v4`, even without direct access to the original physics simulator. Particularly in these tasks, but also in general, we observe severe overfitting happening in CQL almost immediately (for example, CQL performs worse than random in `Pendulum-v1`), likely due to the small size of the provided dataset. As mentioned previously, sample efficiency is one of the main promises of the CWM approach, as very few trajectories are needed to validate the model, whereas traditional methods are typically designed to work best with large amounts of data.

Table 12: **Comparison with CQL.** We report the average raw reward obtained over 10 episodes for a random policy, Conservative Q-Learning (CQL), planning agents with the CWM obtained by GIF-MCTS (ours) respectively with Llama 3 and GPT-4, and a planning agent with oracle access to the true environment (Oracle). CQL was trained with 10 epochs for 100 steps per epoch (1000 total steps) using the *same* dataset used to learn our CWMs.

| Environment | Random | CQL | GIF-MCTS (ours) | | Oracle |
|---|---|---|---|---|---|
| | | | Llama 3 | GPT-4 | |
| Blackjack-v1 | 0 | -0.3 | -0.6 | **-0.1** | 1 |
| CliffWalking-v0 | -1169.2 | N/A* | **-90.2** | -100 | -100 |
| Taxi-v3 | -798.5 | -740 | **-353.9** | -408.8 | -124.5 |
| CartPole-v1 | 24.4 | **317.6** | 277.4 | 310.4 | 494 |
| MountainCar-v0 | **-200** | **-200** | **-200** | **-200** | -200 |
| Acrobot-v1 | -500 | **-295** | -500 | -494.2 | -500 |
| Pendulum-v1 | -1122.8 | -1218.2 | -1232.2 | **-739.8** | -373.6 |
| Reacher-v4 | -43.7 | -11.5 | **-9.2** | -11.2 | -6.8 |
| Pusher-v4 | -149.9 | **-52.4** | -61.1 | -63.3 | -30.3 |
| InvertedPendulum-v4 | 8.3 | **66.7** | 13.1 | 10.9 | 42.5 |
| InvertedDoublePendulum-v4 | 49 | **164** | 60 | 53.4 | 241.6 |
| HalfCheetah-v4 | -304.5 | **-1.3** | -150.3 | -22.8 | 893.3 |
| Hopper-v4 | 32.2 | **137.4** | 62.6 | 23.3 | 229.1 |
| Swimmer-v4 | -5.9 | **28.4** | -2.7 | 8.1 | 317.8 |
| Walker2d-v4 | 0 | **278** | 22.3 | 11.5 | 334.7 |
| Ant-v4 | -33.2 | **998** | 867.7 | 896.8 | 1304.7 |
| Humanoid-v4 | 139.4 | **393.3** | N/A* | 162.3 | 1860.7 |
| HumanoidStandup-v4 | 33240.2 | **51045.7** | N/A* | 29405.9 | 138075.6 |

\* N/A for CQL indicates a failed run, while for GIF-MCTS it indicates a failure in synthesizing a syntactically correct CWM.

It is also worth noting that outperforming state-of-the-art methods for offline RL was not the principal goal we set out to achieve with our work, and as such many aspects are not specifically tuned for performance. For instance, we chose very simple planners with default parameters in order to collect the rewards with the synthesized CWMs, to study the performance of the models in the simplest possible setting. In general, our main objective is to validate the effectiveness of the framework, and we leave improvements that can show increased performance over offline RL methods (for instance, allowing the generated code to call a physics simulator in the continuous environments) to future work, now that the effectiveness of the method has been proven.

## L  Planning algorithms details

In this section we report all the parameters used in our implementations of Monte Carlo Tree Search (MCTS) [Kocsis and Szepesvári, 2006] and Cross Entropy Method (CEM) [Rubinstein, 1997], together with a brief explanation of the meaning of those parameters within the context of the two algorithms.

**MCTS.**  At each time-step, we run $I_{mcts}$ simulations with MCTS to select the best action to play. At every simulation, starting from the root node, we select one action via the Upper-Confidence Bound formula for Trees (UCT)

$$\text{UCT}(\text{node}_i) = v_i + C \cdot \sqrt{\frac{\ln N_i}{n_i + \epsilon}}, \tag{4}$$

where $v_i$ is the estimated value of node $i$, C is the exploration constant, $N_i$ is the visit count of the parent of node $i$, $n_i$ is the visit count of node $i$ and $\epsilon$ is a factor offsetting the visit count. Once we select an unexplored action at one of the nodes, we expand the node that the action leads to

and perform a rollout with a random policy for up to max_actions to estimate its value. The value backpropagation is done as in standard MCTS and we use a discount factor of $\gamma$. The values of all parameters are reported in Table 13.

Table 13: **MCTS planner parameters.**

| Parameter | Description | Value |
|:---:|:---|:---:|
| $I_{mcts}$ | Number of iterations. | 25 |
| max_actions | Max actions per rollout. | 100 |
| $C$ | Exploration constant. | 1.0 |
| $\epsilon$ | Visit count offset. | 1 |
| $\gamma$ | Discount factor. | 0.99 |
| $T_{mcts}$ | Softmax temperature. | 0.01 |

**CEM.** In this case, assuming deterministic environments, we plan directly for the next $T_{cem}$ time-steps, meaning that we choose the actions for up to $T_{cem}$ steps ahead, using the CEM algorithm. At every iteration we sample $N_{cem}$ action plans from a zero-mean Gaussian with dimensions $T_{cem} \times \mathcal{A}$ and standard deviation for each dimension given by half the maximum absolute value between the upper and lower bounds for that action dimension (as it's always the case that each continuous action dimension is bounded in a box in the CWMB environments). The action plans are then clipped in the legal ranges of the action space and scored by their return as rollouts in the environment, starting from the current state. We then select the top $K_{cem}$ action plans (elites samples), fit the Gaussian parameters to them and repeat. At the last iteration, we return the top scoring action plan. All parameters are reported in Table 14.

Table 14: **CEM planner parameters.**

| Parameter | Description | Value |
|:---:|:---|:---:|
| $T_{cem}$ | Time horizon. | 100 |
| $I_{cem}$ | Number of iterations. | 20 |
| $N_{cem}$ | Number of samples. | 1000 |
| $K_{cem}$ | Number of elites. | 100 |

## M  Computational Resources

In the following section we report as accurately as possible the computational resources used in this work. On the high level, the bulk of the computational costs, performed on an AMD cluster, was comprised of the experiments with Llama 3 on APPS, reported in Table 1. The reported experiments require running 3 times Llama 3 on 1000 problems, 20 times each, receiving approximately 1000 tokens in input and producing 1500 tokens in output (as the model is not good in using the End-of-Sequence token to stop earlier). We split the runs in 100 array jobs, each taking approximately 15 hours and requiring 4 AMD MI250x each, for an estimated total of 18000 GPU hours.

Experiments on the CWMB were composed of 18 problems for which we ran our method, one baseline and 3 ablations, which should be roughly equivalent to a single experiment with 100 APPS problems, or 10 jobs of 15 hours with 4 GPUs, for a total of 600 GPU hours. The single experiment performed on RTFM with three different configurations also fits into this budget.

However, many more preliminary attempts were taken, so the full computational budget was of 31.800 GPU hours and a similar amount of CPU hours.

Furthermore, we have paid approximately $62.3 in OpenAI calls to GPT-3.5 Turbo (used only for prototyping) and GPT-4 Turbo (used with a budget of 10 calls on the CWMB experiments in Table 2, with 50 calls in some instances (Table 3) and for other preliminary experiments with GIF-MCTS).

Finally, all environment returns for planning were performed on a single consumer CPU in a few hours.

# N Prompts

In this section we report the main prompts used for GIF-MCTS. These prompts are also shared by our WorldCoder implementation, while we avoid reporting explicitly the prompts used for Zero-shot CoT, as they are simply the problem description followed by "Let's think step by step".

## N.1 APPS Prompts

<system>
You are an experienced Python developer. You will be provided with an incomplete code snippet from a Python program. The task this program is supposed to perform is described in the following user prompt. Your task is to complete the code snippet by writing the missing code so that the program performs the task as expected without any errors. You will be rewarded based on the number of test cases your code passes.
</system>
<user>
{PROB_DESCRIPTION}
Please read the inputs from the standard input (stdin) and print the outputs to the standard output (stdout). Output your code solution with the following format: "'python [your code] "'
</user>
<assistant>
"'python
{CODE_SO_FAR}
</assistant>

Figure 3: Prompt on the APPS benchmark for the *generate* action.

<system>
You are an experienced Python developer. You will be provided with an incorrect code snippet from a Python program. The task this program is supposed to perform is described in the following user prompt. Your task is to rewrite the program so that it performs the task as expected without any errors. You will be rewarded based on the number of test cases your code passes.
</system>
<user>
{PROB_DESCRIPTION}
Please read the inputs from the standard input (stdin) and print the outputs to the standard output (stdout).
First, write an explanation of the difference between the ground-truth output and the program's output in the example provided. Secondly, point out the part of the code responsible for the incorrect prediction and why its logic is erroneous. Third, suggest a concrete, actionable fix for it. Finally fix the program in its entirety following the suggestion. The expected output is in the format:
## Error explanation
[your explanation of the error]
## Error location and wrong logic
[where the error comes from and why]
## Fix suggestion
[how to fix the error]
## Correct code
```python
[your code]
```
## Incorrect code
You are provided with the following code snippet to fix.
```python
{CODE}
```
The code additionally makes a wrong prediction about this input.
## Input
{INPUT}
## Ground-truth output
{OUTPUT}
## Code incorrect outputs
{PREDICTION}
</user>
<assistant>
## Error explanation
</assistant>

Figure 4: Prompt on the APPS benchmark for the *improve* action.

<system>
You are an experienced Python developer. You will be provided with an incorrect Python program. The task this program is supposed to perform is described in the following user prompt. Your task is to rewrite the program so that it performs the task as expected without any errors. You will be rewarded based on the number of test cases your code passes.
</system>
<user>
{PROB_DESCRIPTION}
Please read the inputs from the standard input (stdin) and print the outputs to the standard output (stdout).
First, write an explanation of the error and point out the part of the code responsible for the error and why its logic is erroneous. Second, suggest how you would fix the error, reasoning about the problem. Finally fix the program in its entirety following the suggestion. The expected output is in the format:
## Error explanation
[your explanation of the error]
## Fix suggestion
[how to fix the error]
## Correct code
"'python
[your code]
"'

## Incorrect code
You are provided with the following code snippet to fix.
"'python
{CODE}
"'
{ERROR}
</user>
<assistant>
## Error explanation
</assistant>

Figure 5: Prompt on the APPS benchmark for the *fix* action.

## N.2 CWMB Prompts

<system>
You are an experienced Python developer. You will be provided with an incomplete code snippet from a Python program. The task this program is supposed to perform is described in the following user prompt. Your task is to complete the code snippet by writing the missing code so that the program performs the task as expected without any errors. You will be rewarded based on the number of test cases your code passes.
</system>
<user>
{ENV_DESCRIPTION}
## Class Definition
The class should be called "Environment". It should have at least:
- an __init__ function to set up the Environment, which defines all the variables described in the above documentation, plus any additional variables needed to maintain the environment state or to implement its functionality.
- a set_state function to set a custom value for the environment and its internal representation (you can assume that when "set_state" is used, the task is not done and internal variables should be set as a consequence). set_state takes a single argument as input: a state observation from the observation space defined above.
- a step function to predict a step in the environment. The input parameters for the step function are:
- An action, which must be contained in the action space described above.
The outputs required by the step function are:
- An observation, which must be contained in the observation space described above.
- The reward for taking the action, as described in the reward definition above.
- A boolean variable indicating if the episode is done.
## Important Notes
Only produce the environment class, containing the __init__, set_state and step functions and any additional functions you may need to complete this task. Do not write an example of how to use the class or anything else. Be careful about edge cases. Make sure to write all the required functions and that they have the exact names as specified in the task description. Missing or incorrectly named functions will not pass the tests and will result in a score of 0. It is of VITAL importance that you do not leave undefined any function, but implement each of them completely.
</user>
<assistant>
"'python
{CODE_SO_FAR}
</assistant>

Figure 6: Prompt on the CWMB for the *generate* action.

<system> You are an experienced Python developer. You will be provided with an incorrect code snippet from a Python program. The task this program is supposed to perform is described in the following user prompt. Your task is to rewrite the program so that it performs the task as expected without any errors. You will be rewarded based on the number of test cases your code passes. </system>
<user> {ENV_DESCRIPTION}
## Class Definition
The class should be called "Environment". It should have at least:
- an __init__ function to set up the Environment, which defines all the variables described in the above documentation, plus any additional variables needed to maintain the environment state or to implement its functionality.
- a set_state function to set a custom value for the environment and its internal representation (you can assume that when "set_state" is used, the task is not done and internal variables should be set as a consequence). set_state takes a single argument as input: a state observation from the observation space defined above.
- a step function to predict a step in the environment. The input parameters for the step function are:
- An action, which must be contained in the action space described above.
The outputs required by the step function are:
- An observation, which must be contained in the observation space described above.
- The reward for taking the action, as described in the reward definition above.
- A boolean variable indicating if the episode is done.
## Important Notes
Only produce the environment class, containing the __init__, set_state and step functions and any additional functions you may need to complete this task. Do not write an example of how to use the class or anything else. Be careful about edge cases. Make sure to write all the required functions and that they have the exact names as specified in the task description. Missing or incorrectly named functions will not pass the tests and will result in a score of 0. It is of VITAL importance that you do not leave undefined any function, but implement each of them completely.
First, write an explanation of the difference between the ground-truth transition and the step function's outputs in the example provided. Second, point out the part of the code responsible for the incorrect prediction and why its logic is erroneous. Third, suggest a concrete, actionable fix for it. Finally, fix the program in its entirety following the suggestion. The expected output is in the format:
## Error explanation
[your explanation of the error]
## Error location and wrong logic
[where the error comes from and why]
## Fix suggestion
[how to fix the error]
## Correct code
"'python [your code] "'
## Incorrect code
You are provided with the following code snippet to fix.
"'python {CODE} "'
The code additionally makes a wrong prediction about this input.
## Input
{INPUT}
## Ground-truth output
{OUTPUT}
## Code incorrect outputs
{PREDICTION} </user>
<assistant> ## Error explanation </assistant>

Figure 7: Prompt on the CWMB for the *improve* action.

<system>
You are an experienced Python developer. You will be provided with an incorrect Python program. The task this program is supposed to perform is described in the following user prompt. Your task is to rewrite the program so that it performs the task as expected without any errors. You will be rewarded based on the number of test cases your code passes.
</system>
<user>
{ENV_DESCRIPTION}
## Class Definition
The class should be called "Environment". It should have at least:
- an __init__ function to set up the Environment, which defines all the variables described in the above documentation, plus any additional variables needed to maintain the environment state or to implement its functionality.
- a set_state function to set a custom value for the environment and its internal representation (you can assume that when "set_state" is used, the task is not done and internal variables should be set as a consequence). set_state takes a single argument as input: a state observation from the observation space defined above.
- a step function to predict a step in the environment. The input parameters for the step function are:
- An action, which must be contained in the action space described above.
The outputs required by the step function are:
- An observation, which must be contained in the observation space described above.
- The reward for taking the action, as described in the reward definition above.
- A boolean variable indicating if the episode is done.
## Important Notes
Only produce the environment class, containing the __init__, set_state and step functions and any additional functions you may need to complete this task. Do not write an example of how to use the class or anything else. Be careful about edge cases. Make sure to write all the required functions and that they have the exact names as specified in the task description. Missing or incorrectly named functions will not pass the tests and will result in a score of 0. It is of VITAL importance that you do not leave undefined any function, but implement each of them completely.
First, write an explanation of the error and point out the part of the code responsible for the error and why its logic is erroneous. Second, suggest how you would fix the error, reasoning about the problem. Finally fix the program in its entirety following the suggestion. The expected output is in the format:
## Error explanation
[your explanation of the error]
## Fix suggestion
[how to fix the error]
## Correct code
‘‘‘python
[your code]
‘‘‘
## Incorrect code
You are provided with the following code snippet to fix.
‘‘‘python
{CODE}
‘‘‘
{ERROR}
</user>
<assistant>
## Error explanation
</assistant>

Figure 8: Prompt on the CWMB for the *fix* action.

### N.3 Sample Environment Descriptions

For the CWMB we extract the description for each environment directly from the Gymnasium source code[7]. We clean the description string found for each environment to remove irrelevant information (Arguments, Vectorized Environment, Version History, metadata) as well as manually remove mentions of external links or sources that may provide the LLM with an implementation of the environment. An example description for the CartPole-v1 environment[8] can be seen in Figure 9.

---

## Description
A pole is attached by an un-actuated joint to a cart, which moves along a frictionless track. The pendulum is placed upright on the cart and the goal is to balance the pole by applying forces in the left and right direction on the cart.
## Action Space
The action is a 'ndarray' with shape '(1,)' which can take values '0, 1' indicating the direction of the fixed force the cart is pushed with.
- 0: Push cart to the left - 1: Push cart to the right
**Note**: The velocity that is reduced or increased by the applied force is not fixed and it depends on the angle the pole is pointing. The center of gravity of the pole varies the amount of energy needed to move the cart underneath it
## Observation Space
The observation is a 'ndarray' with shape '(4,)' with the values corresponding to the following positions and velocities:
| Num | Observation | Min | Max |
|——|——————|————————|——————-|
| 0 | Cart Position | -4.8 | 4.8 |
| 1 | Cart Velocity | -Inf | Inf |
| 2 | Pole Angle | -0.418 rad (-24°) | 0.418 rad (24°) |
| 3 | Pole Angular Velocity | -Inf | Inf |
**Note:** While the ranges above denote the possible values for observation space of each element, it is not reflective of the allowed values of the state space in an unterminated episode. Particularly: - The cart x-position (index 0) can be take values between '(-4.8, 4.8)', but the episode terminates if the cart leaves the '(-2.4, 2.4)' range. - The pole angle can be observed between '(-.418, .418)' radians (or **±24°**), but the episode terminates if the pole angle is not in the range '(-.2095, .2095)' (or **±12°**)
## Rewards
Since the goal is to keep the pole upright for as long as possible, a reward of '+1' for every step taken, including the termination step, is allotted. The threshold for rewards is 500 for v1 and 200 for v0.
## Starting State
All observations are assigned a uniformly random value in '(-0.05, 0.05)'
## Episode End
The episode ends if any one of the following occurs:
1. Termination: Pole Angle is greater than ±12° 2. Termination: Cart Position is greater than ±2.4 (center of the cart reaches the edge of the display) 3. Truncation: Episode length is greater than 500 (200 for v0)

Figure 9: Example description for the CartPole-v1 environment.

## O   Examples of Generated Programs

We report examples of Code World Models generated by both Llama 3 and GPT-4 Turbo. We include CartPole-v1 as an example of a discrete environment, Ant-v4 as an example of a continuous environment and RTFM.

---

[7]https://github.com/Farama-Foundation/Gymnasium/tree/main/gymnasium/envs
[8]https://github.com/Farama-Foundation/Gymnasium/blob/main/gymnasium/envs/classic_control/cartpole.py

Figure 10: Code World Model of the CartPole-v1 environment generated by GIF-MCTS with Llama 3. For reference, the official implementation of the environment can be found at the official Gymnasium GitHub repository of the Farama Foundation (at `gymnasium.envs.classic_control.cartpole`).

```python
import numpy as np

class Environment:
    def __init__(self):
        self.cart_position = np.random.uniform(-0.05, 0.05)
        self.cart_velocity = np.random.uniform(-0.05, 0.05)
        self.pole_angle = np.random.uniform(-0.05, 0.05)
        self.pole_angular_velocity = np.random.uniform(-0.05, 0.05)
        self.episode_length = 0
        self.gravity = 9.8
        self.mass_cart = 1.0
        self.mass_pole = 0.1
        self.length_pole = 0.5
        self.force_mag = 10.0
        self.time_step = 0.02  # time step for the simulation

    def set_state(self, state):
        self.cart_position = state[0]
        self.cart_velocity = state[1]
        self.pole_angle = state[2]
        self.pole_angular_velocity = state[3]
        self.episode_length = 0

    def step(self, action):
        force = self.force_mag if action == 1 else -self.force_mag
        costheta = np.cos(self.pole_angle)
        sintheta = np.sin(self.pole_angle)
        temp = (force + self.mass_pole * self.length_pole *
            self.pole_angular_velocity**2 * sintheta) / (self.mass_cart +
            self.mass_pole)
        angular_acceleration = (self.gravity * sintheta - costheta * temp) /
            (self.length_pole * (4/3 - self.mass_pole * costheta**2 /
            (self.mass_cart + self.mass_pole)))
        linear_acceleration = temp - self.mass_pole * self.length_pole *
            angular_acceleration * costheta / (self.mass_cart +
            self.mass_pole)

        # Update the state variables using the calculated accelerations
        new_cart_velocity = self.cart_velocity + linear_acceleration *
            self.time_step
        new_pole_angular_velocity = self.pole_angular_velocity +
            angular_acceleration * self.time_step
        new_cart_position = self.cart_position + self.cart_velocity *
            self.time_step
        new_pole_angle = self.pole_angle + self.pole_angular_velocity *
            self.time_step

        self.cart_velocity = new_cart_velocity
        self.cart_position = new_cart_position
        self.pole_angular_velocity = new_pole_angular_velocity
        self.pole_angle = new_pole_angle

        self.episode_length += 1
        done = abs(self.pole_angle) > 0.2095 or abs(
            self.cart_position) > 2.4 or self.episode_length > 500
        reward = 1
        observation = np.array(
            [self.cart_position, self.cart_velocity, self.pole_angle,
            self.pole_angular_velocity])
        return observation, reward, done
```

Figure 12: Code World Model of the CartPole-v1 environment generated by GIF-MCTS with GPT-4. For reference, the official implementation of the environment * can be found at the official Gymnasium GitHub repository of the Farama Foundation (at `gymnasium.envs.classic_control.cartpole`).

```python
import numpy as np

class Environment:
    def __init__(self):
        self.cart_position = 0.0
        self.cart_velocity = 0.0
        self.pole_angle = 0.0
        self.pole_angular_velocity = 0.0

        self.gravity = 9.8
        self.cart_mass = 1.0
        self.pole_mass = 0.1
        self.total_mass = self.cart_mass + self.pole_mass
        self.pole_length = 0.5  # actually half the pole's length
        self.force_mag = 10.0
        self.tau = 0.02  # time step, 20 ms

        self.reset()

    def reset(self):
        self.cart_position = np.random.uniform(-0.05, 0.05)
        self.cart_velocity = np.random.uniform(-0.05, 0.05)
        self.pole_angle = np.random.uniform(-0.05, 0.05)
        self.pole_angular_velocity = np.random.uniform(-0.05, 0.05)
        return np.array([self.cart_position, self.cart_velocity,
          self.pole_angle, self.pole_angular_velocity])

    def set_state(self, state):
        self.cart_position, self.cart_velocity, self.pole_angle,
          self.pole_angular_velocity = state

    def step(self, action):
        force = self.force_mag if action == 1 else -self.force_mag

        # Equations for the model
        costheta = np.cos(self.pole_angle)
        sintheta = np.sin(self.pole_angle)

        temp = (force + self.pole_mass * self.pole_length *
          self.pole_angular_velocity**2 * sintheta) / self.total_mass
        theta_acc = (self.gravity * sintheta - costheta * temp) /
          (self.pole_length * (4.0/3.0 - self.pole_mass * costheta**2 /
          self.total_mass))
        x_acc = temp - self.pole_mass * self.pole_length * theta_acc *
          costheta / self.total_mass

        # Update the four state variables, using Euler's method
        self.cart_position += self.tau * self.cart_velocity
        self.cart_velocity += self.tau * x_acc
        self.pole_angle += self.tau * self.pole_angular_velocity
        self.pole_angular_velocity += self.tau * theta_acc

        # Check if the episode is done
        done = (
            self.cart_position < -2.4 or self.cart_position > 2.4 or
            self.pole_angle < -0.2095 or self.pole_angle > 0.2095
        )
```

```
53              # Since the problem specifies to always give a reward of +1
54              reward = 1.0
55
56              return np.array([self.cart_position, self.cart_velocity,
                ↪  self.pole_angle, self.pole_angular_velocity]), reward, done
```

Figure 14: Code World Model of the Ant-v4 environment generated by GIF-MCTS with Llama 3. For reference, the official implementation of the environment can be found at the official Gymnasium GitHub repository of the Farama Foundation (at `gymnasium.envs.mujoco.Ant_v4`).

```
1  import numpy as np
2
3
4  class Environment:
5      def __init__(self,
6                       healthy_reward=1.0,
7                       forward_reward_weight=1.0,
8                       ctrl_cost_weight=0.5,
9                       contact_cost_weight=5e-4,
10                      contact_force_range=10.0,
11                      healthy_z_range=(0.2, 1.0),
12                      terminate_when_unhealthy=True,
13                      reset_noise_scale=1e-2,
14                      exclude_current_positions_from_observation=True):
15          self.healthy_reward = healthy_reward
16          self.forward_reward_weight = forward_reward_weight
17          self.ctrl_cost_weight = ctrl_cost_weight
18          self.contact_cost_weight = contact_cost_weight
19          self.contact_force_range = contact_force_range
20          self.healthy_z_range = healthy_z_range
21          self.terminate_when_unhealthy = terminate_when_unhealthy
22          self.reset_noise_scale = reset_noise_scale
23          self.exclude_current_positions_from_observation =
              ↪  exclude_current_positions_from_observation
24          self.state = self._reset_state()
25
26      def _reset_state(self):
27          # implement state reset logic here
28          state = np.zeros(27)
29          state[0] = 0.75   # initial z-coordinate
30          state += np.random.normal(0, self.reset_noise_scale, size=27)
31          return state
32
33      def set_state(self, state):
34          self.state = state
35
36      def step(self, action):
37          # implement dynamics model or physics engine to predict next state
38          next_state = self._predict_next_state(self.state, action)
39          observation = next_state
40          reward = self._calculate_reward(action, next_state)
41          done = self._check_episode_end(next_state)
42          return observation, reward, done
43
44      def _predict_next_state(self, state, action):
45          # implement dynamics model or physics engine to predict next state
46          # for example, using a simple integrator
47          # assume velocities are the last 14 components of the state
48          velocities = state[13:]
49          new_velocities = velocities.copy()
50          # update front left leg, front right leg, back left leg, and back
              ↪  right leg
```

```
51          new_velocities[:4] += 0.01 * action[:4]
52          # update corresponding body parts
53          new_velocities[4:8] += 0.01 * action[4:]
54          # Fix: slice new_velocities to match shape of state[:13]
55          new_positions = state[:13] + 0.01 * new_velocities[:13]
56          next_state = np.concatenate((new_positions, new_velocities))
57          return next_state
58
59      def _calculate_reward(self, action, next_state):
60          # implement reward calculation logic here
61          healthy_reward = self.healthy_reward
62          forward_reward = self.forward_reward_weight * \
63              (next_state[13] - self.state[13])  # x-coordinate velocity
64          ctrl_cost = self.ctrl_cost_weight * np.sum(np.square(action))
65          contact_cost = 0  # implement contact cost calculation logic here
66          reward = healthy_reward + forward_reward - ctrl_cost - contact_cost
67          return reward
68
69      def _check_episode_end(self, state):
70          # implement episode end check logic here
71          done = not (self.healthy_z_range[0] <=
72                      state[0] <= self.healthy_z_range[1])
73          return done
```

Figure 16: Code World Model of the Ant-v4 environment generated by GIF-MCTS with GPT-4. For reference, the official implementation of the environment can be found at the official Gymnasium GitHub repository of the Farama Foundation (at `gymnasium.envs.mujoco.Ant_v4`).

```
1  import numpy as np
2
3  class Environment:
4      def __init__(self, exclude_current_positions_from_observation=True,
        ↪  use_contact_forces=False, terminate_when_unhealthy=True,
5                   healthy_z_range=(0.2, 1.0), healthy_reward=1.0,
                     ↪  ctrl_cost_weight=0.5, contact_cost_weight=0.5,
6                   contact_force_range=(0, 10), reset_noise_scale=0.01,
                     ↪  frame_skip=5):
7          self.exclude_current_positions_from_observation =
           ↪  exclude_current_positions_from_observation
8          self.use_contact_forces = use_contact_forces
9          self.terminate_when_unhealthy = terminate_when_unhealthy
10         self.healthy_z_range = healthy_z_range
11         self.healthy_reward = healthy_reward
12         self.ctrl_cost_weight = ctrl_cost_weight
13         self.contact_cost_weight = contact_cost_weight
14         self.contact_force_range = contact_force_range
15         self.reset_noise_scale = reset_noise_scale
16         self.frame_skip = frame_skip
17         self.dt = self.frame_skip * 0.01
18
19         if self.exclude_current_positions_from_observation:
20             self.observation_space_size = 27
21         else:
22             self.observation_space_size = 29
23
24         if self.use_contact_forces:
25             self.observation_space_size += 84  # 14 body parts * 6 contact
               ↪  force values
26
27         self.state = np.zeros(self.observation_space_size)
28         self.last_x_position = 0.0
29
```

```python
30        def set_state(self, state):
31            self.state = state
32            self.last_x_position = state[0] if not
    ↪   self.exclude_current_positions_from_observation else 0.0
33
34        def step(self, action):
35            if not (-1 <= action).all() or not (action <= 1).all():
36                raise ValueError("Action must be within the range [-1, 1] for each
    ↪   element.")
37
38            # Simulate state transition (simplified)
39            self.state += np.random.normal(0, self.reset_noise_scale,
    ↪   self.observation_space_size)
40
41            x_position = self.state[0] if not
    ↪   self.exclude_current_positions_from_observation else 0.0
42            z_position = self.state[2] if not
    ↪   self.exclude_current_positions_from_observation else self.state[0]
43
44            # Calculate reward components
45            forward_reward = (x_position - self.last_x_position) / self.dt
46            ctrl_cost = self.ctrl_cost_weight * np.sum(np.square(action))
47            contact_cost = 0
48
49            if self.use_contact_forces:
50                # Last 84 entries are contact forces
51                contact_forces = self.state[-84:]
52                clipped_forces = np.clip(contact_forces,
    ↪   self.contact_force_range[0], self.contact_force_range[1])
53                contact_cost = self.contact_cost_weight *
    ↪   np.sum(np.square(clipped_forces))
54
55            reward = self.healthy_reward + forward_reward - ctrl_cost -
    ↪   contact_cost
56            self.last_x_position = x_position
57
58            # Determine if the episode is done
59            unhealthy = not (self.healthy_z_range[0] <= z_position <=
    ↪   self.healthy_z_range[1])
60            done = unhealthy and self.terminate_when_unhealthy
61
62            return self.state, reward, done
63
```

Figure 18: RTFM environment generated by GIF-MCTS with Llama 3

```python
1   import numpy as np
2
3   class Environment:
4       def __init__(self):
5           self.monster_elements = {'demon': 'fire', 'dragon': 'fire', 'jinn':
    ↪   'lightning', 'medusa': 'poison', 'bandit': 'cold', 'wolf': 'cold',
    ↪   'goblin': 'lightning', 'mage': 'lightning', 'spider': 'poison'}
6           self.item_modifiers = {'axe': 'grandmasters', 'bow': 'blessed',
    ↪   'daggers': 'shimmering', 'hammer': 'gleaming', 'polearm':
    ↪   'fanatical', 'shield': 'mysterious', 'staff': 'arcane', 'sword':
    ↪   'soldiers'}
7           self.monster_groups = {'demon': 'rebel enclave', 'dragon': 'order of
    ↪   the forest', 'jinn': 'star alliance', 'medusa': 'order of the
    ↪   forest', 'bandit': 'rebel enclave', 'wolf': 'order of the forest',
    ↪   'goblin': 'star alliance', 'mage': 'star alliance', 'spider':
    ↪   'rebel enclave'}
8           self.goal_group = 'order of the forest'
```

```python
        self.element_weaknesses = {'fire': ['grandmasters', 'soldiers'],
        ↪  'lightning': ['arcane', 'blessed'], 'poison': ['gleaming',
        ↪  'mysterious'], 'cold': ['fanatical', 'shimmering']}
        self.board = np.full((6, 6, 2), 'empty', dtype=object)
        self.board[:, 0, :] = 'wall'
        self.board[:, -1, :] = 'wall'
        self.board[0, :, :] = 'wall'
        self.board[-1, :, :] = 'wall'
        self.inventory = 'empty'
        self.player_position = (1, 1)
        self.board[self.player_position[0], self.player_position[1], 0] =
        ↪  'you'

    def set_state(self, state):
        self.board = state[0]
        self.inventory = state[1]
        for i in range(6):
            for j in range(6):
                if self.board[i, j, 0] == 'you':
                    self.player_position = (i, j)
                    break

    def step(self, action):
        action_dict = {0: "Stay", 1: "Up", 2: "Down", 3: "Left", 4: "Right"}
        new_player_position = list(self.player_position)
        if action_dict[action] == "Up":
            new_player_position[0] -= 1
        elif action_dict[action] == "Down":
            new_player_position[0] += 1
        elif action_dict[action] == "Left":
            new_player_position[1] -= 1
        elif action_dict[action] == "Right":
            new_player_position[1] += 1

        if (0 <= new_player_position[0] < 6 and 0 <= new_player_position[1] <
        ↪  6 and
                self.board[new_player_position[0], new_player_position[1], 0]
                ↪  != 'wall'):
            self.board[self.player_position[0],
                       self.player_position[1], 0] = 'empty'
            self.player_position = tuple(new_player_position)
            self.board[self.player_position[0],
                       self.player_position[1], 0] = 'you'

        frame = (self.board.copy(), self.inventory)
        valid_actions = [i for i in range(5) if self._is_valid_action(i)]
        reward = self._get_reward()
        done = self._is_done()
        return frame, valid_actions, reward, done

    def _is_valid_action(self, action):
        action_dict = {0: "Stay", 1: "Up", 2: "Down", 3: "Left", 4: "Right"}
        new_player_position = list(self.player_position)
        if action_dict[action] == "Up":
            new_player_position[0] -= 1
        elif action_dict[action] == "Down":
            new_player_position[0] += 1
        elif action_dict[action] == "Left":
            new_player_position[1] -= 1
        elif action_dict[action] == "Right":
            new_player_position[1] += 1

        return (0 <= new_player_position[0] < 6 and 0 <=
        ↪  new_player_position[1] < 6 and
```

```
67                    self.board[new_player_position[0], new_player_position[1], 0]
                      ↪  != 'wall')
68
69        def _get_reward(self):
70            for i in range(6):
71                for j in range(6):
72                    if self.board[i, j, 0] in self.monster_elements and
                      ↪  self.monster_groups[self.board[i, j, 0]] ==
                      ↪  self.goal_group:
73                        if self.inventory != 'empty':
74                            item_name = self.inventory.split(' ')[1]
75                            item_modifier = self.item_modifiers[item_name]
76                            monster_element = self.monster_elements[self.board[i,
                          ↪  j, 0]]
77                            if item_modifier in
                          ↪  self.element_weaknesses[monster_element]:
78                                return 1
79                        return -1
80            return 0
81
82        def _is_done(self):
83            for i in range(6):
84                for j in range(6):
85                    if self.board[i, j, 0] in self.monster_elements and
                      ↪  self.monster_groups[self.board[i, j, 0]] ==
                      ↪  self.goal_group:
86                        return False
87            return True
88
```

Figure 20: RTFM environment generated by GIF-MCTS with GPT-4.

```
1   import numpy as np
2
3
4   class Environment:
5       def __init__(self):
6           self.board = np.full((6, 6, 2), 'empty', dtype=object)
7           self.board[:, 0, :] = 'wall'
8           self.board[:, 5, :] = 'wall'
9           self.board[0, :, :] = 'wall'
10          self.board[5, :, :] = 'wall'
11
12          self.player_pos = (3, 1)
13          self.board[3, 1, 0] = 'you'
14          self.inventory = 'empty'
15
16          self.monster_groups = {
17              'star alliance': ['mage', 'goblin', 'jinn'],
18              'order of the forest': ['dragon', 'medusa', 'wolf'],
19              'rebel enclave': ['demon', 'spider', 'bandit']
20          }
21
22          self.monster_elements = {
23              'dragon': 'cold', 'medusa': 'poison', 'wolf': 'poison',
24              'demon': 'fire', 'spider': 'lightning', 'bandit': 'cold',
25              'mage': 'lightning', 'goblin': 'fire', 'jinn': 'cold'
26          }
27
28          self.item_modifiers = {
29              'grandmasters': 'fire', 'blessed': 'lightning', 'shimmering':
                ↪  'cold',
30              'gleaming': 'poison', 'fanatical': 'cold', 'mysterious': 'poison',
```

```
31                'soldiers': 'fire', 'arcane': 'lightning'
32            }
33
34            self.action_dict = {0: "Stay", 1: "Up",
35                                2: "Down", 3: "Left", 4: "Right"}
36            self.goal_group = 'order of the forest'
37
38        def set_state(self, state):
39            board, inventory = state
40            self.board = board
41            self.inventory = inventory
42            pos = np.where(self.board[:, :, 0] == 'you')
43            if pos[0].size > 0:
44                self.player_pos = (pos[0][0], pos[1][0])
45
46        def step(self, action):
47            direction = self.action_dict[action]
48            x, y = self.player_pos
49            dx, dy = x, y
50            if direction == "Up":
51                dx = x - 1
52            elif direction == "Down":
53                dx = x + 1
54            elif direction == "Left":
55                dy = y - 1
56            elif direction == "Right":
57                dy = y + 1
58
59            if self.board[dx, dy, 0] == 'wall':
60                dx, dy = x, y
61
62            new_pos = (dx, dy)
63            cell_content = self.board[new_pos[0], new_pos[1], 0]
64            game_done = False
65            reward = 0
66
67            if cell_content != 'empty' and cell_content != 'wall':
68                content_parts = cell_content.split()
69                if len(content_parts) == 2 and content_parts[1] in
                   ↪  self.monster_elements:
70                    monster = content_parts[1]
71                    monster_element = self.monster_elements[monster]
72                    monster_group = next((group for group, monsters in
                       ↪  self.monster_groups.items(
73                    ) if monster in monsters), None)
74                    if self.inventory != 'empty':
75                        item_modifier, item = self.inventory.split()
76                        if self.item_modifiers[item_modifier] == monster_element
                           ↪  and monster_group == self.goal_group:
77                            reward = 1
78                            game_done = True
79                        else:
80                            reward = -1
81                            game_done = True
82                    else:
83                        reward = -1
84                        game_done = True
85                elif content_parts[0] in self.item_modifiers:
86                    self.inventory = cell_content
87
88            if not game_done:
89                self.board[x, y, 0] = 'empty'
90                self.board[new_pos[0], new_pos[1], 0] = 'you'
91                self.player_pos = new_pos
```

```
92
93          valid_actions = [a for a in self.action_dict if
       ↪   self.board[self.player_pos[0] + (
94              0, -1, 1, 0, 0)[a], self.player_pos[1] + (0, 0, 0, -1, 1)[a], 0]
       ↪   != 'wall']
95
96          return (self.board.copy(), self.inventory), np.array(valid_actions),
       ↪   reward, game_done
```

