# OpenReview forum: "Generating Code World Models with Large Language Models Guided by Monte Carlo Tree Search"
_NeurIPS.cc/2024/Conference — NeurIPS 2024 poster_

### Official Review · Reviewer_49z7 · 2024-07-06

**Soundness:** 2
**Presentation:** 3
**Contribution:** 3
**Rating:** 5
**Confidence:** 5

**Summary:**

This paper explores a novel and promising direction of model-based RL, proposing to represent the dynamic model and the reward model using code world models, which are Python programs that can be executed and rollout environments. With such code world models, we can learn a policy to maximize the return predicted by the models. To achieve this, the paper proposes a framework, Generate, Improve and Fix with Monte Carlo Tree Search (GIF-MCTS), that can synthesize code world models from a pre-collected dataset of environment interactions. This paper also introduces a benchmark for evaluating the performance of code world models. The experiments show that the proposed framework GIF-MCTS can reliably model environments and allow for learning policies that achieve reasonable performance. I believe this work explores an interesting and promising research direction. Yet, I am concerned with the assumptions of offline datasets, the missing comparisons to offline RL methods, and the insufficient description of related work. Hence, I am slightly leaning toward rejection at this moment, but I am willing to increase my score if my concerns are addressed in the author's rebuttal.

**Strengths:**

**Motivation and intuition**
- The motivation for modeling environments using codes is interesting and convincing, and this work presents an effective framework to achieve it.

**Technical contribution**
- Given that WorldCode has explored the online setup, this work explores the offline setup, which offers an unique perspective to the code world model idea.

**Clarity**
- The overall writing is clear. The authors utilize figures well to illustrate the ideas.

**Weaknesses:**

**Dataset collection**
The authors stated that the dataset "includes at least some low-scoring and some relatively high-scoring behavior." I am wondering how we can do this without accessing true reward functions. Given a new environment that we have not seen before, how could we collect such a dataset to learn a code world model?

**The effect of dataset quality**
It is unclear how the quality of the dataset would affect the code world model performance. Can we learn a reasonable code world model with just trajectories collected by a random policy?

**Online setting**
This paper focuses on an offline setup, where the code world model learns from a pre-collected offline dataset. I am curious how we can extend the framework so that it can improve the code world model and the policy constructed from the offline dataset, just like how the research studies how to fine-tune policies learned using offline RL algorithms.

**Access to physics engines as a tool**
The proposed framework writes a code to represent a world model for each environment from scratch. It could be difficult to precisely simulate some interaction that requires modeling physics. It would be interesting to see if granting access to physics engines, e.g., calling MuJoCo, could generalize to physics-intensive environments.

**Comparison to offline RL methods**
This work mainly compares the proposed framework with WorldCoder, which also builds a world model. I am curious about how the proposed framework compares to offline RL methods, which simply learn a policy from the dataset to maximize the return.

**Related work: programmatic RL**
The related work section is not comprehensive. Highly relevant programmatic RL works, which represent policies as programs and interact with environments by executing the programs, are missed from the section, such as:
- Programmatically Interpretable Reinforcement Learning (ICML 2018)
- Imitation-projected programmatic reinforcement learning (NeurIPS 2019)
- Programmatic Reinforcement Learning without Oracles (ICLR 2022)
- Learning to synthesize programs as interpretable and generalizable policies (NeurIPS 2021)
- Hierarchical programmatic reinforcement learning via learning to compose programs (ICML 2023)
- Show me the way! Bilevel search for synthesizing programmatic strategies (AAAI 2023)
- Reclaiming the source of programmatic policies: Programmatic versus latent spaces (ICLR 2024)

**Clarity: offline setup**
When I read the introduction and the abstract, I thought it focused on an online model-based RL setting. Then, I realized this is an offline setup while reading Section 3. I suggest the authors revise the introduction and the abstract to align the expectations.

**Questions:**

See above

**Limitations:**

Yes

---

> ### Author Rebuttal · Authors · 2024-08-07
>
> Thank you for your thoughtful review. We appreciate that you found our motivation compelling and our framework effective. We also appreciate your recognition of our unique perspective on the offline setup of the code world model idea. In the following we address your concerns and we are confident that by including the changes described below the manuscript's quality will be significantly improved.
>
> **1. Dataset collection.** To collect a dataset for a novel environment, one could in practice use a specialised exploration policy, such as in [1], [2] and [3] to mention a few. Including some relatively high-scoring behavior trajectories is common-practice in many offline RL datasets (see e.g., [4]).
> In case no reward is accessible from the environment (if we interpret correctly your question), one could still collect transitions without rewards and learn a partial Code World Model (CWM) that predicts only the next state. In general, dataset quality and exploration for offline RL are active areas of research, for which orthogonal advancements could benefit CWMs.
>
> **2. The effect of dataset quality.** Yes, in principle we can learn a reasonable CWM only with trajectories collected by a random policy. For example in our experiments on RTFM, we employed exclusively a random policy to collect the dataset, which covered the state space sufficiently well and in the best case we obtained a perfect model (Table 3 of the paper).
> If the environment is hard to explore and we are provided only a few random trajectories, it is still be possible to learn a CWM if the language description of the environment is accurate enough.
> We ran an additional experiment on RTFM: we collected 10 trajectories all resulting in failures, so that a reward of +1 is never observed. We synthesized a CWM with GIF-MCTS and GPT-4 using 50 calls. The resulting CWM is 100% accurate on the collected dataset and even correctly predicts a reward of +1 for positive transitions, which are not included in the dataset, thanks to the language description.
>
> **3. Online setting.** This is certainly possible. The simplest procedure is to iteratively **collect data** by interacting with the environment and **update the world model** using the new data, similarly to what is done in the model-based RL literature (see e.g. [5]). To collect data one could use either an exploration policy (point 1) or a variant of the planning algorithms used in our paper with extra noise added for exploration. Our method can be easily modified to update the CWM, for example by re-using the previous tree search and evaluating each program (node) on the updated dataset, before continuing to expand the tree.
>
> In general, we chose to work within the offline setup to focus on studying the overall applicability of the CWM approach without relying on specific exploration approaches that would affect the quality of the training dataset and possibly skew the results. We will add the discussion about dataset collection, dataset quality and extension to online setting to the paper.
>
> **4. Access to physics engines as a tool.** This is most definitely a valid direction for future research and we will mention that. We believe CWMs could be greatly empowered by access to simulation tools. In the specific case of MuJoCo, the simulations are almost fully taken care of by the library, and the resulting CWM would reduce to a trivial function call. Therefore, we feel that to be generalizable to other environments or embodiments, the simulation tool should operate at a more fundamental level.
>
> **5. Comparison to offline RL methods.** We performed a direct comparison with Conservative Q-Learning (CQL), a popular offline RL baseline. We train CQL on the same small dataset of trajectories used for the CWMs. The results are reported in the markdown tables of the common response.
> Overall, there is a balance between CQL and CWMs, with CWMs being more suited to discrete tasks and CQL outperforming CWMs in complex physics tasks. We also observe severe overfitting happening in CQL almost immediately, likely due to the small size of the provided dataset. As we point out in the paper, sample efficiency is one of the main promises of the CWM approach, as very few trajectories are needed to validate the model.
>
> **6. Related work: programmatic RL.** We thank the reviewer for pointing us to this interesting line of research and we will include an additional paragraph about programmatic RL in the Related Work Section. We also added some other works (with [6] and [7] specifically focusing on model-based programmatic RL) and some references for modern works using LLMs to generate programmatic policies.
>
> **7. Clarity: offline setup.** Thanks for the suggestion, we will clarify our setup.
>
> **References:**
>
> [1]: Pathak, Deepak, et al. "Curiosity-driven exploration by self-supervised prediction." International conference on machine learning. PMLR, 2017.
>
> [2]: Savinov, Nikolay, et al. "Episodic Curiosity through Reachability." International Conference on Learning Representations (2019).
>
> [3]: Ecoffet, Adrien, et al. "First return, then explore." Nature 590.7847 (2021): 580-586.
>
> [4]: Fu, Justin, et al. "D4rl: Datasets for deep data-driven reinforcement learning." arXiv preprint arXiv:2004.07219 (2020).
>
> [5]: Hafner, Danijar, et al. "Dream to Control: Learning Behaviors by Latent Imagination." International Conference on Learning Representations.
>
> [6]: Azad, Abdus Salam, et al. "Scenic4rl: Programmatic modeling and generation of reinforcement learning environments." arXiv preprint arXiv:2106.10365 (2021).
>
> [7]: Tsividis, Pedro A., et al. "Human-level reinforcement learning through theory-based modeling, exploration, and planning." arXiv preprint arXiv:2107.12544 (2021).

---

> > ### Comment · Reviewer_49z7 · 2024-08-09
> > **Re: Rebuttal by Authors**
> >
> > Thank you for the rebuttal, which addresses some of my concerns. I am increasing my score to 5 (borderline accept), counting on the authors will revise the paper according to the promises, e.g., being upfront about the offline setup in the abstract and the introduction, including the comparisons to offline RL methods, etc.

---

> > > ### Author Response · Authors · 2024-08-11
> > >
> > > Thank you for your thoughtful consideration and for increasing your score. We are committed to revising the paper as promised, ensuring that the offline setup is clearly stated in the abstract and introduction, and including the necessary comparisons to offline RL methods. We appreciate your feedback and will make the necessary improvements.

---

### Official Review · Reviewer_3vW3 · 2024-07-09

**Soundness:** 2
**Presentation:** 1
**Contribution:** 2
**Rating:** 6
**Confidence:** 3

**Summary:**

The authors present a search algorithm based on Monte Carlo tree search to synthesize programs with LLMs. The contribution includes formulating a search space that is compatible with the functionalities of a LLM: the actions in the search tree include generating new lines of code, fixing bugs, and improving current implementation.

The system is evaluated on the synthesis of programs for solving competition-level problems and programs encoding world models. As baselines, the experiments include recent works such as CodeRL, Parsel, and WorldCoder.

**Strengths:**

After reading the paper once, I believe its main contribution is to show the power of tree search in the context of synthesis with LLMs. A simple MCTS-based algorithm already produces programs able to attain better quality metrics than other methods not using search.

The experiments are nicely done from an ablation perspective: I quite enjoyed Table 6 in the appendix, where different versions of GIF-MCTS are evaluated.

**Weaknesses:**

The paper has 4 weaknesses that are worth mentioning, so the authors can react to them in the rebuttal.

**1. Clarity of the paper could be much improved.**

When reading Figure 2 and the description around it, I got confused with the use of the word "rollout." I think it is being used to mean two different things. The first is a portion of the code that a node represents. The second is the LLM call that generates that portion.

I did not find in the paper the specification in terms of how many lines are generated in each "generate new lines" action and how many are generated in the rollouts.

**2. The paper overclaims**

> we improve the trade-off between exploration and exploitation in the UCT formula used for action selection.

What was probably meant was that the paper uses a heuristic that achieves better empirical results. There are no theoretical improvements in the trade-off.

> Calling code instead of LLMs for planning has the advantages of being precise, reliable, interpretable, and extremely efficient.

None of these properties are actually properly evaluated in the paper. This might be a bit of nitpicking, but code can also be imprecise, unreliable, uninterpretable, and extremely inefficient. Anyone working as a programmer has already experienced all these negative properties of code.

**3. Lack of discussion on data contamination**

With the exception of the RTFM benchmark, nothing is said or discussed about possible data contamination. Is the search algorithm helping the LLM remember what is in its training data or is the overall solution doing something else? Are the models for RL tasks available online as code? Are the solutions to the programming problems available online?

The lack of discussion about data contamination brings me to the question of what is the research question behind this work. Is it a retrieval question (i.e., can the system recover information used in the LLM's training data) or a reasoning question (i.e., can the system write programs that solve problems)?

**4. Statistical significance is unclear**

It is not clear whether the search was run only once or many times. The averages presented seem to be over a number of problems and episodes, but not on the number of times the entire system was executed.

**Questions:**

1. Please explain the issue around the use of the word "rollout" and how the number of lines of code is determined in each call of the "generate new lines" actions.
2. How about running the fix bug action a number of times whenever a state representing a buggy program is generated? Instead of allowing MCTS to decide when to debug, why not debug right away?
3. Please comment on the data contamination issue mentioned above and clarify what is the research question behind this work.
4. Please comment on the statistical metrics and empirical design of the experiments.

**Limitations:**

In weaknesses, please see the parts related to overclaiming and data contamination.

---

> ### Author Rebuttal · Authors · 2024-08-07
>
> Thank you for the thorough review and the valuable suggestions to improve the clarity and soundness of our work. We appreciate your recognition of the power of GIF-MCTS and of the usefulness of the ablation studies on our method. We detail in the following how we are going to address your comments in the final version of our work.
>
> **1. Clarity.**
>
> **Rollout.** We consider each node as a full program, comprised of a "state" part and a "rollout" part. The **state** is the **first $L \times d$ lines**, where $L$ is the number of generated new lines per node and $d$ is the depth of a node in the tree. The **remaining part of the program** is denoted as the **rollout**, and we consistently use the term to mean only that. **LLM calls** correspond to actions (edges) in the tree, and each of them **produces a full program** (node).
> Please let us know if there is a specific part of the manuscript that creates this confusion and we will clarify it in the final version.
>
> **Generated lines.** The number of generated new lines $L$ is equal to 2 (Table 4 in the Appendix); we will report this information also in the main text.
> We also realised that in the main text we used $l$ rather than $L$ and we will fix this.
>
> **2. Claims.** We never meant to present imprecise claims and we thank the reviewer for pointing out the need for better wording of them.
> > we improve the trade-off between exploration and exploitation in the UCT formula used for action selection.
>
> We will change to "we *propose a heuristic that empirically improves* the trade-off between exploration and exploitation in the UCT formula used for action selection" (justified in Appendix A).
>
> > Calling code instead of LLMs for planning has the advantages of being precise, reliable, interpretable, and extremely efficient.
>
> We will change to "Calling code instead of LLMs for planning has *potential* to be *more* precise, reliable, interpretable, and extremely efficient."
>
> Addressing the mentioned properties one by one:
> - Precise: code can natively access a calculator and as such is numerically precise.
> - Reliable: a program passing all unit tests can be deemed more reliable than a black-box prediction from an LLM.
> - Interpretable: as mentioned in our response with reviewer u7M2, we will add concrete examples of Code World Models (CWMs) generated by our method, as mentioned to Reviewer u7M2. We include one example on Ant-v4 in the common response. In general, we find that all generated programs are clearly interpretable by a proficient human programmer.
> - Extremely efficient: we report in Table 7 of the appendix (referred in the Introduction section) the inference time of our generated CWMs vs inference time of an LLM. **CWMs are 4 to 6 orders of magnitude faster.**
>
> **3. Data contamination.**
>
> **RTFM Benchmark:** The RTFM benchmark is not available online, and our method's success on it provides evidence that our solution is not merely retrieving information from the LLM's training data.
>
> **Programming Problems:** The programming problems we used are sourced from three main websites. The benchmark authors managed to crawl reference solutions for only two of these sites. Performance across all methods and models in the competition split is correlated with the source websites of the problems, but not with the availability of the solutions: the highest results are obtained from Kattis, the only site where solutions are not available online. Notably, all methods and models achieve a 0% pass rate for the 41 problems from AtCoder, for which reference solutions are available online.
>
> **Gym Environments:** While we observe that some parts of the gym environments recall implementations available online (e.g., constants' values in the CartPole environment), the logic of the step function remains distinct from the reference model. We include one model-generated environment in the common response, and we will add more in the final manuscript, along with links to the most popular online implementations.
>
> **Fair Comparison:** Data contamination is a known issue in almost all studies involving LLMs, not just ours. However, our method is compared to baselines using the same underlying models, ensuring that its superior performance is not biased by potential data contamination.
>
> We will add this discussion to the Appendix of the paper.
>
> **4. Statistical significance.** The search was run once per each problem, due to the computational cost. For APPS, considering that we have 1000 independent problems, this is more than enough to tell apart with statistical significance the results of GIF-MCTS and the baselines. For CWMB, the uncertainity is higher. We ran 2 extra seeds for each environment (resulting in 18 environments * 3 seeds total executions for our method) for GIF-MCTS and WorldCoder using Llama 3 70B and obtain the following results:
>
> | Model                | Method          | Budget | Discrete Accuracy     | Discrete Return        | Continuous Accuracy      | Continuous Return      |
> |----------------------|-----------------|---------|---------------|---------------|---------------|---------------|
> | Llama 3 70B| GIF-MCTS (ours) | 50      | **0.84±0.03** | **0.76±0.03** | **0.35±0.03** | **0.22±0.01** |
> | Llama 3 70B | WorldCoder    | 50      | 0.79±0.04     | 0.60±0.04     | 0.32±0.03     | 0.19±0.01     |
>
> CWM still outperforms WorldCoder with smaller error margins, confirming the statistical significance of our method, especially in the discrete case. We unfortunately could not repeat this for GPT-4 due to budget concerns.
>
> **5. When to fix bugs.** Great question. Sometimes, generating a bug-free program from scratch will be more promising than trying to fix a buggy program multiple times, whereas other times the opposite will be true. Rather than deciding a priori how many times to debug right away, we believe that letting MCTS decide will lead to lower regret. A further relevant discussion is present in Appendix A.

---

> > ### Comment · Reviewer_3vW3 · 2024-08-12
> >
> > Thank you for clarifying the issues raised in my review.
> >
> > I have increased the overall score of my review.

---

### Official Review · Reviewer_u7M2 · 2024-07-13

**Soundness:** 3
**Presentation:** 4
**Contribution:** 3
**Rating:** 8
**Confidence:** 3

**Summary:**

This paper primarily investigates the application of LLMs to synthesizing world models for reinforcement learning environments, where the world model is expressed as a Python program implementing the state transition and reward functions. Concretely, starting from available environment simulation code (e.g., a benchmark environment using the MuJoCo physics simulator), the authors first obtain a number of trajectories. An off-the-shelf LLM is then prompted with to reimplement the environment, starting from a textual description (and without explicit access to the original code), and the collected trajectories are used as test cases to verify correctness. This is a challenging code synthesis problem for which the authors propose an iterative algorithm to refine and improve the generated program with multiple LLM calls, formulated as monte-carlo tree search.

To me, the contributions of this paper are:
- The GIF-MCTS method for effective iterative program synthesis wrt available test cases, which is separately evaluated on the APPS code generation benchmark.
- The concept of code world models (as outlined above) which shows promise in efficiently obtaining world models from observational data
- A benchmark which covers common environments used in Deep RL that enables the community to propose further advances

**Strengths:**

I found the main idea of the paper to be quite original and interesting, i.e., generating the code for a world model rather than utilizing LLMs directly. The authors show that this works quite well in environments with discrete action space: they often obtain perfect or near-perfect prediction accuracy as well as rewards approaching those achieved by planning with an oracle world model. In continuous environments, where the original implementations rely on a physics simulation engine, the results are less convincing -- this is understandable to me as I would expect LLMs (or humans, for that matter) having trouble reimplementing those without access to the original or a substitute simulation engine. Whether code world models are a promising direction to ultimately tackle planning in complex or real-life environments remains to be seen, of course; however, I would expect paper at hand and the proposed benchmark to inspire further works in this relevant direction.

I also liked the separate evaluation of their MCTS code synthesis method on APPS where it produces strong results, and it's worth noting that there are full papers proposing similar iterative prompting for code generation. I hope the authors make this part of the paper available for easy re-use.

The paper itself is well-written and, while it is pretty packed, I found that I could easily locate and understand the method and details on problem setup and evaluations.

**Weaknesses:**

I would appreciate if the authors included examples of LLM-generated environment implementations in the appendix. I would be interested in how the continuous action space environments are being tackled, or to check for overlap with the original environment implementations (which

For the planning results it would be good to list the true rewards obtained along with the rewards of a random policy and SOTA model-free/model-based results from the RL literature. While beating SOTA results here is not the focus of the paper, it would provide a helpful perspective for readers familiar with those environments.

**Questions:**

I am not totally clear on the LLM sample budget for the evaluation on APPS. From L258 ff: "B is the budget for the number of LLM calls", Table 1 lists pass@20, but L618 in the appendix mentions 50 Llama 3 calls per problem. L618 also mentions 1000 problems, whereas the competition split on which you report results is smaller.

**Limitations:**

I found the paper's discussion of its limitations to be upfront and comprehensive and have nothing to add.

---

> ### Author Rebuttal · Authors · 2024-08-07
>
> Thank you for the kind words about our work. We are glad that you found it interesting and believe it will inspire future works! We agree that continuous physics simulations are particularly challenging and we also were not particularly surprised by that, but we will be looking to address this in future work, for example by integrating the method with external physics simulators and APIs.
>
> **1. Examples of LLM-generated environments:**
> >I would appreciate if the authors included examples of LLM-generated environment implementations in the appendix. I would be interested in how the continuous action space environments are being tackled, or to check for overlap with the original environment implementations
>
> In the limited space of the rebuttal, we include a Code World Model (CWM) example for the Ant environment (continuous action and state spaces) in the common response.
> In the final manuscript, we will include two more (one gym environment with discrete action space and RTFM), together with how they compare to the ground truth source code.
> We performed an additional qualitative study on the generated programs and found that while certain aspects of the gym environments resemble available online implementations (such as the values of constants in the CartPole environment), the logic of the step function is distinct from that of the reference model for all generations.
>
> **2. Comparison with RL:**
> > It would be good to list the true rewards obtained along with the rewards of a random policy and SOTA model-free/model-based results from the RL literature.
>
> We performed a direct comparison to a random baseline, the oracle baseline (using the same planner as the CWMs but with the ground truth environment) and with Conservative Q-Learning [1], a popular offline RL baseline that can work for both discrete and continuous action spaces. We train CQL on the same small dataset of trajectories used for the CWMs. The results are reported in the markdown tables of the common response.
> Overall, there is a balance between CQL and CWMs, with CWMs being more suited to discrete tasks and CQL outperforming CWMs in complex physics tasks. We also observe severe overfitting happening in CQL almost immediately, likely due to the small size of the provided dataset. As we point out in the paper, sample efficiency is one of the main promises of the CWM approach, as very few trajectories are needed to validate the model. There is also a considerable gap between CQL and the oracle planner, which represents an upper bound for the CWM approach given our choice of planners.
>
> **3. LLM sample budget:**
> > I am not totally clear on the LLM sample budget for the evaluation on APPS. From L258 ff: "B is the budget for the number of LLM calls", Table 1 lists pass@20, but L618 in the appendix mentions 50 Llama 3 calls per problem. L618 also mentions 1000 problems, whereas the competition split on which you report results is smaller.
>
> The mention of 50 calls in the Appendix was a typo, thank you for catching it! Indeed, the correct budget used for APPS is 20 LLM calls corresponding to 20 programs generated and evaluated on the unit tests. As for the number of problems, we work with the test set of the APPS dataset which is composed of 5000 total problems, of which 1000 make up the competition split, so 1000 is the correct number of problems we evaluated on. Could you point us to where you got the impression that the competition split is smaller?
>
> **References:**
>
> [1]: Kumar, Aviral, et al. "Conservative q-learning for offline reinforcement learning." Advances in Neural Information Processing Systems 33 (2020): 1179-1191.

---

### Official Review · Reviewer_WBDc · 2024-07-13

**Soundness:** 3
**Presentation:** 3
**Contribution:** 2
**Rating:** 6
**Confidence:** 4

**Summary:**

This paper proposes Generate, Improve, and Fix with Monte Carlo Tree Search (GIF-MCTS) for generating Code World Models (CWMs) using Large Language Models (LLMs). The authors present code representations of reinforcement learning (RL) environments, enabling the application of LLM algorithms for code generation across various domains. They introduce the Code World Models Benchmark (CWMB), which comprises 18 diverse RL environments to evaluate their approach. GIF-MCTS demonstrates better performance compared to the baselines of APPS, CWMB, and a grid-world environment.

**Strengths:**

The paper is well-written and effectively demonstrates that MCTS can facilitate agentic behaviors and problem-solving across multiple domains. The motivation and the rationale are clear, as we expect the idea of agentic behavior of searching with diverse actions can help solve more complex problems.

The appendix includes extensive ablative studies that offer a detailed analysis of the method's components.

**Weaknesses:**

**Reliance on pre-training.** I didn’t fully understand why learning the world model is necessary. MCTS is an inference-time algorithm. The reliance on pre-training for the large language models might be a limitation.

Specifically, does GIF-MCTS need to predict all transitions well in the offline dataset to perform well on a task? For coding, the agent doesn’t need to accurately predict how a human solves a problem. It’s sufficient if the agent can find a correct solution, possibly using MCTS.

**Novelty.** There seems to be limited novelty in this framework. Related works have explored agentic behaviors of using “implementing a function”, “fixing bug” as actions, like SWE-agent [1]. Using tree search and sequential decision-making are also explored in the literature [2]. The novelty seems to be a combination of both.

References

[1] Yang, John, et al. "Swe-agent: Agent-computer interfaces enable automated software engineering." arXiv preprint arXiv:2405.15793 (2024).

[2] Zhang, Shun, et al. "Planning with large language models for code generation." arXiv preprint arXiv:2303.05510 (2023).

**Questions:**

Can the authors clarify why GIF-MCTS is better than WorldCoder? Is it because of the advantage of the tree search algorithm (while WorldCoder is mainly a sampling algorithm)?

Offline pre-training seems to be expensive. For domains like code generation, is pre-training necessary? For new domains that the agent may not have seen in the pre-training data, does it help to use in-context learning to show the agent some example trajectories?

**Limitations:**

The limitations are clearly discussed in the paper.

---

> ### Author Rebuttal · Authors · 2024-08-07
>
> Thank you for taking the time to review our work and provide your detailed feedback. We are grateful for your comments and we hope that the following will clarify the details and novelty of our work.
>
> **1. Pre-training:**
> > Reliance on pre-training. I didn’t fully understand why learning the world model is necessary. MCTS is an inference-time algorithm. The reliance on pre-training for the large language models might be a limitation.
>
> and
>
> > Offline pre-training seems to be expensive. For domains like code generation, is pre-training necessary? For new domains that the agent may not have seen in the pre-training data, does it help to use in-context learning to show the agent some example trajectories?
>
> We would like to clarify a potential misunderstanding regarding our approach. We do not train nor fine-tune any LLM, but only use off-the-shelf LLMs (e.g., Llama-3, GPT-4).
> Specifically, we learn a Code World Model (CWM) that can be used in model-based RL as a substitute for the dynamics and reward functions of the environment, allowing an agent to search over action strategies using methods such as CEM or standard MCTS.
> Separately, we propose GIF-MCTS, an iterative generation method tailored for generating CWMs with LLMs. Within it, we make use of in-context learning to show trajectories to the model as you suggested, specifically with the improve action. We developed GIF-MCTS specifically because we found prompting the LLM directly to not be effective.
>
> **2. Predicting transitions:**
> >Specifically, does GIF-MCTS need to predict all transitions well in the offline dataset to perform well on a task?
>
> In our work, we assume this is the case. More precisely, GIF-MCTS produces the CWM, which in turn should predict correctly all transitions in the offline dataset to perform well as a world model. However, there may be cases where the model does not need to be uniformly accurate everywhere, particularly in suboptimal regions rarely visited by the planning policy.
>
> > For coding, the agent doesn’t need to accurately predict how a human solves a problem. It’s sufficient if the agent can find a correct solution, possibly using MCTS.
>
> It is not fully clear to us what the reviewer means by this. Our work involves two agents: the GIF-MCTS one, whose objective is to write a correct CWM, and the model-based RL agent, which uses vanilla MCTS (or CEM) with the CWM to solve the RL task (e.g., CartPole). Neither of the two ever tries to predict how a human solves a problem and no human data is used to learn the world model. For coding, the reward function used by GIF-MCTS (which guides the generation process) is the fraction of passed unit tests-- no full solutions are compared against.
>
> **3. Novelty:**
> > Novelty. There seems to be limited novelty in this framework. Related works have explored agentic behaviors of using “implementing a function”, “fixing bug” as actions, like SWE-agent [1]. Using tree search and sequential decision-making are also explored in the literature [2]. The novelty seems to be a combination of both.
>
> We respectfully disagree. Our main novelty lies in formulating the concept of Code World Models, a framework to learn RL world models written in code. Additionally, we introduce a benchmark to evaluate this approach, and finally propose GIF-MCTS as a code-generation algorithm with an LLM, specifically tailored for the CWM framework.
> Hence, while GIF-MCTS can be seen as novel combination of agentic behaviour and tree search, our novelty goes beyond that.
>
> **4. Comparison with WorldCoder:**
> > Can the authors clarify why GIF-MCTS is better than WorldCoder? Is it because of the advantage of the tree search algorithm (while WorldCoder is mainly a sampling algorithm)?
>
> Good point, thank you for raising it. We believe that GIF-MCTS outperforms WorldCoder because it considers a more diverse set of programs.
> The main difference is that WorldCoder initially generates a single complete program, which becomes the ancestor for all future programs. In contrast, GIF-MCTS can generate multiple programs either from scratch or from partial programs by taking the "generate new lines" action at the root node or subsequent nodes, which better explores the solution space.
> A further ablation study ("No Generate action" in Table 6 of the Appendix) supports this finding: using a tree search (like GIF-MCTS) but only with refinement actions (like WorldCoder) results in lower performance compared to our method.
> We will add this explanation to the Discussion section in the main text.

---

> ### Comment · Reviewer_WBDc · 2024-08-11
> **Thanks for the response**
>
> Thank you for the responses and clarifications. The clarification on pre-training does clear up my confusion. I thought the training data set $D$ (line 148) is used to train a language model to generate world models. That was a misunderstanding.
>
> I wanted to make sure thatm I am following the logic of the paper correctly. Could you confirm if the following points are accurate?
>
> - GIF-MCTS is a new tree search algorithm evaluated on the APPS dataset, where it outperforms baseline algorithms like WorldCoder.
> - GIF-MCTS is then applied to Code World Model learning and evaluated on the Code World Models Benchmark and Read to Fight Monsters. The results show that GIF-MCTS learns a better world model than WorldCoder, and planning with the GIF-MCTS-learned world models achieves better returns.
>
> If these points are correct, it would be helpful to clarify them in the paper. Initially, I thought we needed to learn a world model for APPS, which confused me.

---

> > ### Author Response · Authors · 2024-08-11
> >
> > Thank you for your careful review and for taking the time to clarify your understanding. We are glad that the confusion was cleared away and we can confirm that your points are accurate.
> >
> > We will clarify that GIF-MCTS is a tree search-based code generation algorithm and that we use APPS to evaluate its performance on the general code synthesis task, but that APPS and CWMs are a related but separate application and that the method is framed differently for APPS (for example the dataset D becomes the collection of unit tests).
> >
> > We hope that this helps to explain the novelty of the paper: in our view, the central contribution is the concept of a Code World Model and the possible use cases for RL, while GIF-MCTS is a tree search algorithm we designed specifically for the task of synthesizing better CWMs, with the evaluation on APPS serving the purpose of generally evaluating the method on a widely used benchmark.
> >
> > Please let us know if there are any concerns that remained unaddressed and thank you again for your time and effort.

---

> ### Comment · Reviewer_WBDc · 2024-08-11
> **Thanks for your response**
>
> Thanks for confirming that my understanding is correct. Overall I believe learning a world model represented by code is an interesting and promising idea, which is also validated empirically. I believe the paper would benefit from improved clarity and the new results discussed in the rebuttal.
>
> I have raised the score to 6 (Weak Accept).

---

### Author Rebuttal · Authors · 2024-08-07

We thank all the reviewers for engaging with our work and providing their precious feedback. In addition to our individual answers, we include the following material:

**Example of generated Code World Model.** Reviewers u7M2 and 3vW3 asked for examples of Code World Models (CWMs), being interested in the generated world models for continuous environments and in possible overlaps with the online implementations. In Figure 1 of the attached PDF we report the CWM generated by our method for the Ant-v4 environment. For reference, the official implementation of the environment can be found at the official Gymnasium GitHub repository of the Farama Foundation (at gymnasium.envs.mujoco.Ant_v4).

**Comparison with RL baselines.** Reviewer u7M2 asked for reference a comparison with SOTA RL and Oracle baselines, and reporting the raw returns, while reviewer 49z7 was specifically interested in a comparison with offline RL. We report in the following tables the average reward obtained over 10 episodes for a random policy, a SOTA method Conservative Q-Learning (CQL) [1], planning agents with the CWM obtained by GIF-MCTS (ours) respectively with Llama 3 and GPT-4, and a planning agent with oracle access to the true environment (Oracle). CQL is a SOTA offline RL method, trained with 10 epochs for 100 steps per epoch (1000 total) using the *same* dataset used to learn our CWMs. We chose 1000 steps to match the data to gradient steps ratio from the original CQL paper. Since our replay buffers are much smaller, we started to observe severe overfitting for CQL with more training steps.

| **Environment (Discrete)**              | **Random** | **CQL**  | **GIF-MCTS (ours) Llama 3** | **GIF-MCTS (ours) GPT-4** | **Oracle** |
|------------------------------|------------|----------|-----------------------------|---------------------------|------------|
| Blackjack-v1              |      0   |    -0.3 |      -0.6 |    **-0.1** |      1   |
| CliffWalking-v0           |  -1169.2 |   N/A   |     **-90.2** |  -100   |   -100   |
| Taxi-v3                   |   -798.5 |  -740   |    **-353.9** |  -408.8 |   -124.5 |
| CartPole-v1               |     24.4 |   **317.6** |     277.4 |   310.4 |    494   |
| MountainCar-v0            |   **-200**   |  **-200**   |    **-200**   |  **-200**   |   -200   |
| Acrobot-v1                |   -500   |  **-295**   |    -500   |  -494.2 |   -500   |


---

| **Environment (Continuous)**              | **Random** | **CQL**  | **GIF-MCTS (ours) Llama 3** | **GIF-MCTS (ours) GPT-4** | **Oracle** |
|------------------------------|------------|----------|-----------------------------|---------------------------|------------|
| Pendulum-v1               |  -1122.8 | -1218.2 |   -1232.2 |  **-739.8** |   -373.6 |
| Reacher-v4                |    -43.7 |   -11.5 |      **-9.2** |   -11.2 |     -6.8 |
| Pusher-v4                 |   -149.9 |   **-52.4** |     -61.1 |   -63.3 |    -30.3 |
| InvertedPendulum-v4       |      8.3 |    **66.7** |      13.1 |    10.9 |     42.5 |
| InvertedDoublePendulum-v4 |     49   |   **164**   |      60   |    53.4 |    241.6 |
| HalfCheetah-v4            |   -304.5 |    **-1.3** |    -150.3 |   -22.8 |    893.3 |
| Hopper-v4                 |     32.2 |   **137.4** |      62.6 |    23.3 |    229.1 |
| Swimmer-v4                |     -5.9 |    **28.4** |      -2.7 |     8.1 |    317.8 |
| Walker2d-v4               |     -0   |   **278**   |      22.3 |    11.5 |    334.7 |
| Ant-v4                    |    -33.2 |   **998**   |     867.7 |   896.8 |   1304.7 |
| Humanoid-v4               |    139.4 |   **393.3** |     N/A   |   162.3 |   1860.7 |
| HumanoidStandup-v4        |  33240.2 | **51045.7** |     N/A   | 29405.9 | 138076   |

N/A for CQL indicates a failed run, while for GIF-MCTS indicates a failure in synthesizing a syntactically correct CWM.

In general, planning with the generated CWMs works best on discrete environments, where our original results also indicate that the CWMs are of a higher quality. However, CWMs also reach competitive results in some complex physics tasks, such as Pendulum-v1, Reacher-v4 and to a lesser extent Ant-v4, Pusher-v4 and HalfCheetah-v4, without direct access to the original physics simulator.

**References:**

[1]: Kumar, Aviral, et al. "Conservative q-learning for offline reinforcement learning." Advances in Neural Information Processing Systems 33 (2020): 1179-1191.

---

### Decision · Program_Chairs · 2024-09-25

**Decision:**

Accept (poster)

**Comment:**

Many thanks to the authors for submitting this work and engaging in discussion with reviewers. I appreciate that the authors introduced an innovative MCTS approach for using LLM as an agent in an RL-based setup. Specifically, the authors introduced Code World Models to learn RL world models formulated as code/programs to facilitate LLM agents. The authors reported comprehensive experimental results and ablation analysis to demonstrate the effectiveness of the proposed method. The authors also contributed a new benchmark, including applicable environments for optimizing and evaluating RL agents.

As noted by reviewers, there are some minor issues: (1) the tasks for evaluating code world models are limited to simple domains; it is still an opening question of how effectively Code World Models might work in more complex and practical scenarios.  (2) secondly, some presentation issues should be addressed e.g. Figure 2 and the offline setup,  and claims that need to be adjusted (as pointed out by Reviewer 3vW3).

In conclusion, I recommend accepting this paper. I encourage the authors to review all the above-mentioned points and other feedback from reviewers and revise their paper accordingly.